# TEST-TIME VERIFICATION VIA OPTIMAL TRANSPORT: COVERAGE, ROC, & SUB-OPTIMALITY

**Arpan Mukherjee[†], Marcello Bullo[†], Debabrota Basu[\*], Deniz Gündüz[†]**

[†]Imperial College London
`{a.mukherjee,m.bullo21,d.gunduz}@imperial.ac.uk`

[\*]Equipe School, Université de Lille, Inria, CNRS
`{debabrota.basu}@inria.fr`

## ABSTRACT

While test-time scaling with verification has shown promise in improving the performance of large language models (LLMs), role of the verifier and its imperfections remain underexplored. The effect of verification manifests through interactions of three quantities: (i) the generator's *coverage*, (ii) the verifier's *region of convergence* (ROC), and (iii) the sampling algorithm's *sub-optimality*. Though recent studies capture subsets of these factors, a unified framework quantifying the geometry of their interplay is missing. We frame verifiable test-time scaling as a transport problem. This characterizes the interaction of coverage, ROC, and sub-optimality, and uncovers that the sub-optimality–coverage curve exhibits three regimes. A *transport regime* – where sub-optimality increases with coverage, a *policy improvement regime* – where sub-optimality may decrease with coverage, depending on the verifier's ROC, and a *saturation regime* – where sub-optimality plateaus, unaffected by coverage. We further propose and analyze two classes of sampling algorithms – *sequential* and *batched*, and examine how their computational complexities shape these trade-offs. Empirical results with `Qwen`, `Llama`, and `Gemma` models corroborate our theoretical findings.

## 1 MOTIVATIONS & CONTRIBUTIONS

Test-time scaling has emerged as a promising axis for improving the performance of large language models (LLMs) (Jaech et al., 2024). Existing approaches for test-time scaling are generally categorized as verifier-free and verifier-based (details in Appendix A). Verifier-based methods leverage verifiers — binary reward mechanisms grounded in *de facto* correctness criteria (e.g., unit tests, gold solutions). These verifiers have widely shown potential to improve post-training performance while used in both training and inference phases (Cobbe et al., 2021; Guo et al., 2025; Luo et al., 2025; Huang et al., 2025a; Dorner et al., 2025).

Typical verifier-based test-time pipelines consist of a generator (the reference LLM), a verifier, and a sampling algorithm (e.g., Best-of-$N$ (BoN) (Aminian et al., 2025)). Performance of the generated responses (e.g., accuracy for objective tasks) results from the combined attributes of each of these components. Following rapid empirical progress, efforts have been made to uncover the theoretical underpinnings of test-time verification, specifically its aggregate scaling laws such as $pass@N$ performance (Brown et al., 2024) and policy divergence (Beirami et al., 2024). A majority of these studies assume an *accurate* verifier, a simplifying assumption which is seldom satisfied in practice. While recent studies investigate these imperfections (Huang et al., 2025a; Dorner et al., 2025), a unified perspective that elucidates the interactions between the components' characteristics and verification inaccuracy is missing. Motivated by this gap, we ask the following overarching question:

> To what extent can verifier-based sampling approximate the induced optimal policy, and how are the approximations shaped by verification inaccuracies?

Addressing these questions requires moving beyond the asymptotic scaling curves, and towards a finer-grained analysis that captures the exact dependence of performance on the generator, the ver-

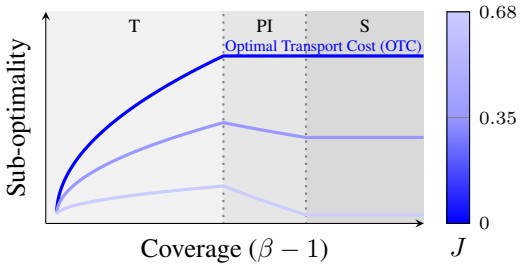

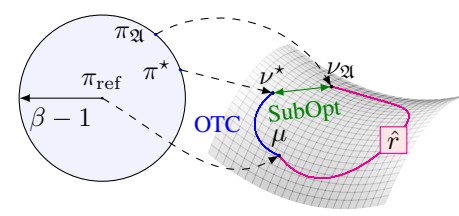

Figure 1: Regimes of test-time verification.      Figure 2: Geometry of test-time verification.

ifier, and the sampling algorithm. In this paper, we study the interplay between the generator's *coverage*, the verifier's *region of convergence* (ROC), and the sampling algorithm's *sub-optimality* through an exact analysis. We formulate test-time verification as a sampling problem. Given generative access to a proposal distribution $\mu$, we are tasked with sampling from a target distribution $\nu^\star$. The only access we have to the target distribution is through an approximately correct verifier $\widehat{r}$, assuming a *de facto* ground truth $r^\star$. In this context, we make the following contributions:

**I. Framework.** By recognizing test-time verification as a sampling problem, we study it through the lens of optimal transport. Here, the goal is to transport the proposal distribution $\mu$ of the reference LLM to a target distribution $\nu^\star$, defined by the ground-truth verifier $r^\star$. Since $\nu^\star$ is not directly accessible, we instead rely on discriminative access via an imperfect verifier $\widehat{r}$ to guide sampling. If the algorithm accepts proposals too generously, the induced distribution remains close to $\mu$, leading to high sub-optimality. Conversely, if it applies an overly stringent rejection policy and discards most proposals, sub-optimality may shrink, albeit at the cost of an excessive compute budget. The key challenge is to design a transport plan that balances proposal usage against induced sub-optimality.

**II. Geometry of sub-optimality vs. coverage.** We decompose sub-optimality into (i) an optimal transport cost, capturing the intrinsic difficulty of transporting the proposal distribution $\mu$ to the target $\nu^\star$, and (ii) a policy improvement term, reflecting how the sampling algorithm mitigates this cost. Sampling directly from $\nu^\star$ achieves policy improvement exactly matching the transport cost, and yielding an optimal sampling scheme. In practice, however, verifier inaccuracies render this ideal infeasible, and sub-optimality is governed jointly by the verifier's ROC, particularly, its Youden's index, and the generator's coverage. Our analysis reveals that as coverage constraints are relaxed, the sub-optimality$-$coverage curve exhibits three distinct regimes, as depicted in Figure 1:

1. *Transport regime*, where the optimal transport cost dominates policy improvement;
2. *Policy improvement regime*, where the optimal transport cost saturates and a sufficiently accurate verifier enables sub-optimality reduction;
3. *Saturation regime*, where both terms plateau, leaving sub-optimality constant regardless of further coverage.

**III. Algorithms and their properties.** We study two protocols: a sequential generation protocol, where responses are generated until acceptance, and a batched generation protocol, where a batch of responses is drawn, and the algorithm distills a winning response. In the sequential protocol, we revisit the naïve *accept-if-correct* (AiC) strategy analyzed by Dorner et al. (2025)[1] and show that AiC violates our coverage constraint in the transport regime. To address this limitation, we propose *sequential rejection sampling* (SRS), a valid transport plan for which we derive exact sub-optimality. In addition, to reduce the number of proposals, we introduce *sequential maximal coupling* (SMC), which minimizes transport cost and achieves the same sub-optimality as SRS. Surprisingly, despite being derived from different principles, SRS and SMC require the same expected number of proposals. Table 1 summarizes the properties of AiC, SRS, and SMC. Finally, to account for batched generation schemes such as BoN sampling, we investigate batched variants of SRS and BoN. Our analyses and empirical studies[2] reveal that rejection sampling-type algorithms are better suited to low-coverage regimes, whereas BoN-type algorithms are preferable under relaxed coverage.

---

[1]Dorner et al. (2025) refer to this strategy as rejection sampling. Our analysis, however, distinguishes rejection sampling from AiC, motivating our separate nomenclature.

[2]The code is available at https://github.com/marcellobullo/ot-resampling.

| Metrics | Sequential | | | Batched | |
| --- | --- | --- | --- | --- | --- |
| | AiC | SRS | SMC | BoN | BRS |
| Coverage | PI, S | T, PI, S | T, PI, S | PI, S | T, PI, S |
| Computational complexity | $\frac{1}{s_{\text{ver}}}$ (Thm. 3.2) | $\frac{(1 \wedge m(s_{\text{ver}}))}{s_{\text{ver}}}$ (Thm. 3.5) | $\frac{(1 \wedge m(s_{\text{ver}}))}{s_{\text{ver}}}$ (Thm. 3.5) | $N+1$ | $N+1$ |
| Sub-optimality | $\text{OTC}\,(1 - \tilde{\alpha}_k\,J)$ (Thm. 3.2) | $\text{OTC}\,(1 - \alpha_k\,J)$ (Thm. 3.6) | $\text{OTC}\,(1 - \alpha_k\,J)$ (Thm. 3.6) | Thm. O.1 | Thm. O.2 |

Table 1: Complexity and sub-optimality across algorithms. OTC is the optimal transport cost, $s_{\text{ver}}$ is the generator's mass on the verifier set, $m(s_{\text{ver}})$ is the induced optimal policy's mass on that set, $J$ is Youden's index, $k \in \{\text{T}, \text{PI}, \text{S}\}$, with T = transport, PI = policy improvement, S = saturation.

## 2 FORMULATION: TEST-TIME VERIFICATION AS A TRANSPORT PLAN

Let $\mathcal{X}$ be the space of prompts.[3] Each prompt $\mathbf{x} \in \mathcal{X}$ admits a response $\mathbf{y} \in \mathcal{Y}$ generated by a reference LLM with conditional kernel $\pi_{\text{ref}}(\cdot \mid \mathbf{x})$. For generality, we assume $\mathcal{Y}$ is a Polish space equipped with a Borel $\sigma$-algebra $\mathfrak{B}(\mathcal{Y})$. The induced reference distribution over responses is $\mu \triangleq \text{law}(\mathbf{Y} \mid \mathbf{X} = \mathbf{x})$. Test-time verification assumes existence of a ground-truth verifier $r^\star : \mathcal{X} \times \mathcal{Y} \mapsto \{0, 1\}$ that assigns a binary reward to each (prompt, response) pair. Specifically, we model verification as a set-membership problem, i.e., for each prompt $\mathbf{x} \in \mathcal{X}$, there exists a set of correct responses $\mathcal{S}^\star(\mathbf{x}) \subseteq \mathcal{Y}$, and the verifier asserts membership via $r^\star(\mathbf{x}, \mathbf{y}) \triangleq \mathbb{1}\{\mathbf{y} \in \mathcal{S}^\star(\mathbf{x})\}$. $\mathcal{S}^\star(\mathbf{x})$ abstracts different verifier designs depending on the task. For example, in a coding problem, $\mathcal{S}^\star(\mathbf{x})$ corresponds to all programs that pass the unit tests. For a math problem, it represents all solutions that yield a correct final answer, possibly attained by different reasoning steps or expressed in different yet mathematically equivalent forms. When using LLM-as-a-judge, $\mathcal{S}^\star(\mathbf{x})$ contains the set of responses with scores exceeding a predetermined threshold characterizing the de facto ground truth. Since all notations implicitly depend on the prompt $\mathbf{x}$, we omit this dependency for brevity whenever it is unambiguous from the context.

**Coverage and optimal policy.** Test-time verification is a sampling problem, where the goal is to sample from a target distribution that maximizes the average reward obtained from the verifier. However, it is unrealistic to define an optimal policy that may arbitrarily deviate from the generator, since responses from such an optimal policy might not be generatable via sampling from the reference policy. Hence, following the state-of-the-art (Huang et al., 2025a), we adopt an $\ell_1$-type coverage constraint on the class of optimal policies. Specifically, we constrain the optimal policy to belong to a set of policies, which are *sufficiently covered* by the reference LLM, i.e.,

$$\Pi(\beta \mid \mathbf{x}) \triangleq \left\{ \pi(\cdot \mid \mathbf{x}) : \mathcal{X} \mapsto \Delta(\mathcal{Y}) \,\Big|\, \mathbb{E}_{\mathbf{Y} \sim \pi(\cdot \mid \mathbf{x})}\left[ \frac{\pi(\mathbf{Y} \mid \mathbf{x})}{\pi_{\text{ref}}(\mathbf{Y} \mid \mathbf{x})} \right] \leq \beta \right\}, \tag{1}$$

where $\Delta(\mathcal{Y})$ denotes the space of Borel probability measures on $\mathcal{Y}$. Note that (1) implies that $\chi^2(\mu\|\nu) \leq \beta - 1$ for any measure $\nu$ (induced by $\pi$), where $\chi^2(\mu\|\nu) \triangleq \int_{\mathcal{Y}} (\frac{d\nu}{d\mu})^2 \mu(d\mathbf{y}) - 1$ denotes the $\chi^2$-divergence between measures $\mu$ and $\nu$[4]. We overload the notation $\Pi(\beta \mid \mathbf{x})$ to denote both the set of conditional kernels and the set of induced probability measures satisfying the constraint. Hence, the optimal policy is $\pi^\star(\cdot \mid \mathbf{x}) \in \arg\sup_{\pi \in \Pi(\beta \mid \mathbf{x})} \mathbb{E}_{\mathbf{y} \sim \pi(\cdot \mid \mathbf{x})}[r^\star(\mathbf{x}, \mathbf{y})]$, and the corresponding measure is $\nu^\star \triangleq \text{law}(\mathbf{Z} \mid \mathbf{X} = \mathbf{x})$, where $\mathbf{Z} \sim \pi^\star(\cdot \mid \mathbf{x})$. Next, leveraging the binary structure of the verifier's reward, we obtain a closed-form expression for the induced optimal policy.

**Theorem 2.1** (Analytical Form of Optimal Policy). *For any (prompt, response) pair* $(\mathbf{x}, \mathbf{y}) \in \mathcal{X} \times \mathcal{Y}$, *let* $r(\mathbf{x}, \mathbf{y}) \triangleq \mathbb{1}\{\mathbf{y} \in \mathcal{S}_r(\mathbf{x})\}$ *be a verifier, where* $\mathcal{S}_r(\mathbf{x}) \subseteq \mathcal{Y}$. *Further, let* $\nu_r \triangleq \arg\sup_{\nu \in \Pi(\beta \mid \mathbf{x})} \int r(\mathbf{x}, \mathbf{y}) \, d\nu(\mathbf{y}|\mathbf{x})$ *denote the induced optimal measure. The Radon-Nikodym*

---

[3]We use $\mathbf{Z}$, $\mathbf{z}$, and $\mathcal{Z}$ to denote a random vector, its realization, and a set, respectively.

[4]Note that the $\chi^2$-divergence is more stringent, as it upper bounds several commonly used divergences, including the order-2 Rényi divergence, the 1-Wasserstein distance, and the Kullback–Leibler divergence. Consequently, if a constraint holds in $\chi^2$-divergence, it also holds for each of these divergences.

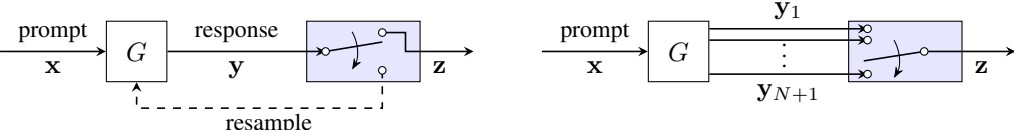

Figure 3: Sequential (*left*) and batched (*right*) sampling protocols of test-time verification with a generator $G$ and a verifier (light purple box).

*derivative of the target measure $\nu_r$ with respect to the reference $\mu$, denoted by $\eta_r \triangleq \frac{\mathrm{d}\nu_r}{\mathrm{d}\mu}$, is*

$$\eta_r(\mathbf{y}) \triangleq \begin{cases} \left(\frac{1}{s_r} \wedge \frac{m_\beta(s_r)}{s_r}\right) & \text{if } \mathbf{y} \in \mathcal{S}_r, \\ \left(0 \vee \frac{1-m_\beta(s_r)}{1-s_r}\right) & \text{if } \mathbf{y} \notin \mathcal{S}_r, \end{cases} \tag{2}$$

*where $m_\beta(s) \triangleq s + \sqrt{s(1-s)(\beta-1)}$, and $s_r \triangleq \int_{\mathcal{S}_r} r \, \mathrm{d}\mu$.*

**Sampling as a transport problem.** We now cast sampling as a transport problem. Given generative access to $\mu$, the goal of a sampling algorithm is to obtain samples from $\nu^\star$ by formalizing a valid transport plan that is a coupling between $\mu$ and $\nu^\star$. We define the set of all couplings $\mathcal{M}(\mu, \nu)$ between the reference $\mu$ and any target $\nu$ as the set of all joint measures on $\mathcal{Y} \times \mathcal{Y}$ such that its projections on the first and second coordinates are $\mu$ and $\nu$, respectively. For any coupling $\rho(\mathrm{d}\mathbf{y}, \mathrm{d}\mathbf{z}) \in \mathcal{M}(\mu, \nu)$, we assign the Hamming cost as the price to be paid for transporting $\mu$ to $\nu$ through $\rho$, i.e., $C(\rho) \triangleq \int_{\mathcal{Y} \times \mathcal{Y}} \mathbb{1}\{\mathbf{y} \neq \mathbf{z}\}\rho(\mathrm{d}\mathbf{y}, \mathrm{d}\mathbf{z})$. The average Hamming cost captures the fraction of rejections required to sample from the target $\nu$, and hence, comes up as a natural candidate for the transport cost. A sampling algorithm $\mathfrak{A}$ is characterized by a coupling $\rho_\mathfrak{A}(\mathrm{d}\mathbf{y}, \mathrm{d}\mathbf{z})$ such that its projection on the first coordinate yields the reference law $\mu$, and we define $\nu_\mathfrak{A}$ as the projection of $\rho$ on the second coordinate. In order to capture the efficiency of the sampling algorithm in generating from the optimal policy $\nu^\star$, we adopt the notion of *sub-optimality* that assesses the change in policy performance of RL pre- and post-training (Zhu et al., 2023; Huang et al., 2025a):

$$\mathrm{SubOpt}(\mathfrak{A}) \triangleq \int r^\star(\mathbf{x}, \mathbf{y}) \, \mathrm{d}\nu^\star(\mathbf{y} \mid \mathbf{x}) - \int r^\star(\mathbf{x}, \mathbf{y}) \, \mathrm{d}\nu_\mathfrak{A}(\mathbf{y} \mid \mathbf{x}). \tag{3}$$

It immediately follows that any transport plan $\rho(\mathrm{d}\mathbf{y}, \mathrm{d}\mathbf{z}) \in \mathcal{M}(\mu, \nu^\star)$ is optimal. The challenge, as depicted in Figure 2, is that we do not have access to $\nu^\star$, but only membership access to an *approximately correct verifier* $\widehat{r} : \mathcal{Y} \times \mathcal{Y} \mapsto \{0, 1\}$, such that for any response $\mathbf{Y} \sim \pi_{\mathrm{ref}}(\cdot \mid \mathbf{x})$ generated for a prompt $\mathbf{x} \in \mathcal{X}$, an approximately correct reward signal $\widehat{r}(\mathbf{x}, \mathbf{y}) = \mathbb{1}\{\mathbf{y} \notin \widehat{\mathcal{S}}\}$ is available to the sampling algorithm for some $\widehat{\mathcal{S}} \subseteq \mathcal{Y}$. The optimal policy, which lies within a $\chi^2$-ball of radius $\beta - 1$ from $\pi_{\mathrm{ref}}$, induces the maximal reward on the manifold induced by $r^\star$. Given access to $\widehat{r}$, the sampling algorithm's distribution $\nu_\mathfrak{A}$ should also satisfy the $\chi^2$-constraint. Naturally, the approximation quality of the verifier should affect the sampling performance. To formalize this, we define the true positive rate (TPR), false positive rate (FPR), and Youden's index $J$ (Youden, 1950) of the imperfect verifier as

$$\mathrm{TPR} \triangleq \frac{1}{s_{r^\star}}\mu\big(\widehat{\mathcal{S}} \cap \mathcal{S}\big), \quad \mathrm{FPR} \triangleq \frac{1}{1 - s_{r^\star}}\mu\big(\widehat{\mathcal{S}} \setminus \mathcal{S}\big), \quad \text{and} \quad J \triangleq \mathrm{TPR} - \mathrm{FPR}.$$

These are the standard quantifiers of the goodness of binary classifiers (Kumari & Srivastava, 2017; Santos et al., 2019), or equivalently, the power of binary hypothesis tests (Li & Tong, 2020).

## 3 ALGORITHMS & ANALYSIS: SEQUENTIAL AND BATCHED SAMPLING

We study different test-time verification algorithms as transport plans, and analyze their properties. Depending on how the sampling algorithm interacts with the generator, we study the *sequential* and *batched* protocols, as illustrated in Figure 3 and detailed below:

- *Sequential sampling protocol:* Generation is modeled as a sequential decision process. Given a prompt $\mathbf{x} \in \mathcal{X}$, at each round $n \in \mathbb{N}$, the generator produces a response $\mathbf{Y}_n \sim \pi_{\mathrm{ref}}(\cdot \mid \mathbf{x})$. The

sampling algorithm $\mathfrak{A}$ observes the history $Y^n \triangleq (\mathbf{Y}_1, \ldots, \mathbf{Y}_n) \in \mathcal{Y}^n$, and based on verifier feedback, issues a decision $\delta_n^{\mathfrak{A}} : \mathcal{Y}^n \to \{\text{accept}, \text{reject}\}$. If a response is accepted, the algorithm stops and outputs it. Otherwise, it queries the generator for another sample. The instant at which $\tau_{\mathfrak{A}}$ stops sampling is referred as the *stopping time*, $\tau_{\mathfrak{A}} \triangleq \inf \{ n \in \mathbb{N} : \delta_n^{\mathfrak{A}}(Y^n) = \text{accept} \}$.

- *Batched sampling protocol:* In the batched setting, following Huang et al. (2025a), the generator produces $N + 1$ independent responses in parallel. The sampling algorithm inspects any $N$ of them, using the verifier to identify a candidate. If none are accepted, the algorithm defaults to returning *any one* of the $(N + 1)$ responses.

To evaluate sampling algorithms, we consider two key metrics: the number of proposals (computational efficiency), and the algorithm's sub-optimality (performance efficiency). In the sequential setting, the computational efficiency is measured by the expected number of proposals $\mathbb{E}[\tau_{\mathfrak{A}}]$, while the sub-optimality is defined as in Equation (3). There is a natural tension between these objectives: drawing more samples may reduce sub-optimality, albeit at the expense of larger computation. In the batched setting, the computational budget is fixed at $N+1$ samples. Hence, performance is evaluated solely through sub-optimality. In what follows, we introduce a range of sampling algorithms under both protocols and analyze their performance with respect to these metrics.

### 3.1 SEQUENTIAL SAMPLING ALGORITHMS: AiC, SRS, AND SMC

We present three algorithms for sequential protocol: (1) a naïve AiC algorithm asserting membership of the generated response through the approximate verifier, (2) an SRS algorithm which has a mechanism of accepting a sample even if its set-membership assertion fails, and (3) an SMC algorithm derived by optimizing the transport cost. For brevity, we defer the pseudo-codes to Appendix B.

Central to analyzing these algorithms is the optimal (Hamming) transport cost (OTC), which is the minimum probability of rejections required to transport the reference law $\mu$ to the target $\nu^\star$, and is defined as $\text{OTC} \triangleq \min_{\rho \in \mathcal{M}(\mu, \nu^\star)} C(\rho)$. The following lemma provides a closed-form for OTC.

**Lemma 3.1** (Optimal Transport Cost (OTC) for Hamming distance). *Given Hamming cost* $c(\mathbf{y}, \mathbf{z}) \triangleq \mathbb{1}\{\mathbf{y} \neq \mathbf{z}\}$, *the OTC for Hamming distance is* $\text{OTC}(\beta) = (1 \wedge m_\beta(s_{r^\star})) - s_{r^\star}$.

#### 3.1.1 ACCEPT-IF-CORRECT (AiC)

The AiC algorithm (Algorithm 1), proposed by (Dorner et al., 2025) is an extension of the BoN sampling strategy to the sequential setting. Given the (approximate) verifier through the set-membership oracle $\widehat{S}$, at each time $n \in \mathbb{N}$, AiC samples a response $\mathbf{Y}_n \sim \mu$, asserts its membership in $\widehat{S}$, and resamples if the assertion fails. Noticeably, AiC ignores the policy coverage bound $\beta$, resulting in constraint violation in certain coverage regimes. In the following theorem, we characterize the average number of proposals and sub-optimality of AiC.

**Theorem 3.2** (Computational complexity and sub-optimality of AiC). *Let* $s_{\text{ver}} \triangleq s_{r^\star}\text{TPR} + (1 - s_{r^\star})\text{FPR}$ *denote the verifier acceptance probability. The expected computational complexity of AiC is* $\mathbb{E}[\tau_{\text{AiC}}] = 1/s_{\text{ver}}$, *and its sub-optimality is given by*

$$\text{SubOpt}(\text{AiC}) = \begin{cases} \text{OTC}(\beta) \cdot \left(1 - \frac{s_{r^\star}}{s_{\text{ver}}} \cdot J\right), & \text{if } \beta > \frac{1}{s_{r^\star}}, \\ \text{OTC}(\beta) \cdot \left(1 - \frac{1}{s_{\text{ver}}} \sqrt{\frac{s_{r^\star}(1-s_{r^\star})}{\beta-1}} \cdot J\right), & \text{if } \beta \leq \frac{1}{s_{r^\star}}. \end{cases}$$

Theorem 3.2 shows that AiC's sub-optimality depends linearly on (a) the OTC, and (b) Youden's index $J$. A smaller Youden's index, corresponding to a verifier closer to random guessing, yields higher sub-optimality. Since AiC ignores the coverage constraint, its sub-optimality is not comparable uniformly over all coverage regimes. As evident from the next theorem, AiC fails to satisfy the coverage requirement in low-coverage regimes, thereby incurring constraint violations.

**Theorem 3.3** (Constraint violation of AiC). *Given any prompt* $\mathbf{x} \in \mathcal{X}$, *AiC policy* $\pi_{\text{AiC}}(\cdot \mid \mathbf{x})$ *does not satisfy the coverage constraint for* $\beta < 1/s_{\text{ver}}$, *i.e.,* $\pi_{\text{AiC}}(\cdot \mid \mathbf{x}) \notin \Pi(\beta \mid \mathbf{x})$ *for all* $\beta < 1/s_{\text{ver}}$.

### 3.1.2 SEQUENTIAL REJECTION SAMPLING (SRS)

To circumvent AiC's lack of coverage, we propose a rejection sampling (RS)-based algorithm (Algorithm 2), which is aware of the coverage constraint.

Canonical RS (Forsythe, 1972; Neal, 2003) evaluates a scaled likelihood ratio against a uniform random variable to determine sample acceptance, essentially flipping a Bernoulli coin where the scaling factor, known as the envelope, dictates the acceptance probability. However, our context lacks the target-to-proposal likelihood ratio $\eta_{r^\star}$, since $\mathcal{S}^\star$ is unknown and the sampling algorithm only has access to an approximate membership oracle $\widehat{\mathcal{S}}$. We therefore introduce SRS, which substitutes $\eta_{\widehat{r}}$, obtained by replacing $s_{r^\star}$ in Equation (2) with $s_{\widehat{r}}$. While $s_{\widehat{r}}$ is computable in principle, it may not be accessible at test time. In Section 4, we treat $s$ as a tunable hyperparameter with ablations across multiple models. Our theoretical analyses, however, assume that $s_{\widehat{r}}$, the reference policy's probability mass on the verifier's set $\widehat{\mathcal{S}}$, is available to the sampling algorithm. While seemingly strong, this assumption enables a fundamental characterization of how the verifier's ROC influences sub-optimality. Note that by our construction, SRS *always* satisfies the coverage constraint. Performance analysis of SRS is presented jointly with our next algorithm, SMC, to facilitate direct comparison and maintain brevity.

### 3.1.3 SEQUENTIAL MAXIMAL COUPLING (SMC)

Maximal coupling (MC) (Algorithm 3) is a canonical technique for constructing optimal transport maps (Den Hollander, 2012). The goal is to find a joint distribution $\rho^\star$ that *minimizes* the transport cost, i.e., $\rho^\star \in \arg\min_{\rho \in \mathcal{M}(\mu, \nu^\star)} C(\rho)$. Under the Hamming cost, this amounts to minimizing the rejection probability, suggesting the potential to improve computational efficiency. The MC algorithm in this setting is well studied: the generator first produces a sample, which is evaluated by the sampling algorithm. The algorithm compares the likelihood ratio at this sample against a uniform random draw. If the ratio exceeds the threshold, the sample is accepted. Otherwise, MC samples from a *residual measure* as a correction. Consequently, MC requires at most two proposals to produce a valid sample from the target distribution.

In the test-time setting, however, the sampling algorithm lacks access to samples from the residual measure, making a direct application of MC infeasible. Nevertheless, we identify an alternative representation of the residual that is generatable, as formalized in the following lemma.

**Lemma 3.4** (Residual measure). *Given a proposal measure $\mu$ and a target measure $\nu$ on $\mathcal{Y}$ induced by a verifiable reward $r$ with a membership oracle $\mathcal{S}$, the residual distribution for MC, defined as $\mu_{\mathrm{res}} \triangleq (\nu - (\mu \wedge \nu))/(1 - (\mu \wedge \nu)(\mathcal{Y}))$, can be equivalently characterized as $\mu_{\mathrm{res}} = \mu(\cdot \mid \mathcal{S})$, where we have defined the conditional measure $\mu(\cdot \mid \mathcal{S}) \triangleq \mu(\cdot \cap \mathcal{S})/\mu(\mathcal{S})$.*

Leveraging Lemma 3.4, we now extend canonical MC to a sequential protocol, and propose SMC. We start similarly to MC, i.e., drawing a response and a uniform number, and then comparing the likelihood ratio of the obtained sample to the uniform random realization. If the ratio exceeds the uniform number, SMC accepts the sample. Otherwise, SMC keeps drawing samples from $\mu$ until the generated sample asserts the set-membership verification rather than sampling from the residual. Evidently, not having access to a residual measure incurs a computational price to mimic sampling from the target measure. In the following theorem, we characterize the computational complexities of both SRS and SMC algorithms, and find that they require the *same* average number of proposals.

**Theorem 3.5** (Computational complexity of SRS and SMC). *Let $M \triangleq \max\left\{\left(\frac{1}{s_{\mathrm{ver}}} \wedge \frac{m_\beta(s_{\mathrm{ver}})}{s_{\mathrm{ver}}}\right),\right.$ $\left.\left(0 \vee \frac{1-m_\beta(s_{\mathrm{ver}})}{1-s_{\mathrm{ver}}}\right)\right\}$ for SRS. For both algorithms $\mathfrak{A} \in \{\mathrm{SRS}, \mathrm{SMC}\}$, the computational complexity is identical, and given by $\mathbb{E}[\tau_{\mathfrak{A}}] = \frac{1}{s_{\mathrm{ver}}}\left(1 \wedge m_\beta(s_{\mathrm{ver}})\right)$.*

Note that the computational complexity of SRS and SMC improves upon AiC by a factor of $m_\beta(s_{\mathrm{ver}})$. Under liberal coverage constraints, where $m_\beta(s_{\mathrm{ver}}) = 1$, their complexity coincides with that of AiC. In contrast, under more stringent coverage, SRS and SMC achieve a computational speed-up over AiC. Next, we provide SRS and SMC sub-optimality, and find that both sub-optimalities follow a piecewise curve divided into three distinct regimes.

**Theorem 3.6** (Sub-optimality of SRS & SMC). *Sub-optimalities of SRS and SMC are expressed as*
$$\mathrm{SubOpt}(\mathfrak{A}) = \mathrm{OTC}(\beta) \cdot (1 - \alpha J),$$
*where $\mathfrak{A} \in \{\mathrm{SRS}, \mathrm{SMC}\}$, and $\alpha$ varies depending on the coverage constraint $\beta$ as follows:*

1. Transport regime $\beta \le \left(\frac{1}{s_{r^\star}} \wedge \frac{1}{s_{\mathrm{ver}}}\right)$: we have $\alpha = \sqrt{\frac{s_{r^\star}(1-s_{r^\star})}{s_{\mathrm{ver}}(1-s_{\mathrm{ver}})}}$;

2. Policy Improvement regime $\left(\frac{1}{s_{r^\star}} \wedge \frac{1}{s_{\mathrm{ver}}}\right) \le \beta \le \left(\frac{1}{s_{r^\star}} \vee \frac{1}{s_{\mathrm{ver}}}\right)$: we have

$$\alpha = \begin{cases} \frac{1}{s_{\mathrm{ver}}}\sqrt{\frac{s_{r^\star}(1-s_{r^\star})}{\beta-1}} & \text{if } s_{\mathrm{ver}} > s_{r^\star}, \beta \in \left(\frac{1}{s_{\mathrm{ver}}}, \frac{1}{s_{r^\star}}\right], \\ s_{r^\star}\sqrt{\frac{\beta-1}{s_{\mathrm{ver}}(1-s_{\mathrm{ver}})}} & \text{if } s_{\mathrm{ver}} \le s_{r^\star}, \beta \in \left(\frac{1}{s_{r^\star}}, \frac{1}{s_{\mathrm{ver}}}\right]; \end{cases}$$

3. Saturation regime $\beta > \left(\frac{1}{s_{r^\star}} \vee \frac{1}{s_{\mathrm{ver}}}\right)$: we have $\alpha = \frac{s_{r^\star}}{s_{\mathrm{ver}}}$.

Theorem 3.6 reveals three distinct regimes. In the transport regime, sub-optimality grows as $O(\sqrt{\beta})$ and is fully governed by $\mathrm{OTC}(\beta)$. In the policy improvement regime, if $s_{\mathrm{ver}} \le s_{r^\star}$ and Youden's index is positive, the policy reduces sub-optimality. By contrast, $s_{\mathrm{ver}} \ge s_{r^\star}$ admits false positives and yields no improvement. In the saturation regime, $\mathrm{OTC}(\beta)$ stabilizes at $1 - s_{r^\star}$, and hence, sub-optimality remains constant despite increasing coverage.

Theorems 3.5 and 3.6 collectively establish that SMC, despite its design, is no more computationally efficient than SRS, as the lack of residual access offsets potential gains. Thus, SRS and SMC exhibit *equivalent performance* in both computational complexity and sub-optimality. Theorems 3.2 and 3.3 show that AiC violates constraints in the transport regime, while matches SRS and SMC in the saturation regime, supporting their use under liberal coverage. Finally, while Huang et al. (2025a) report sub-optimality scaling with the square root of coverage, our analysis refines this observation: *the trade-off between the coverage and the sub-optimality is not universal but mediated by the verifier's ROC.*

## 3.2 Batched Sampling Algorithms: BoN and BRS

In this section, we analyze BoN, characterizing its sub-optimality and identifying the maximal batch size $N+1$ beyond which constraint violations occur. We then introduce a batched variant of rejection sampling (BRS), and establish that it satisfies coverage constraints for *all* batch sizes. For our analysis, we focus on accurate verifiers; extension to approximately correct verifiers is deferred to Appendix O.

### 3.2.1 Best-of-N (BoN)

Given a prompt $\mathbf{x} \in \mathcal{X}$, BoN obtains independent and identically distributed (iid) responses $\mathbf{y}^{N+1} \triangleq (\mathbf{y}_1, \cdots, \mathbf{y}_{N+1})$ from the proposal $\mu$. Subsequently, it returns a response $\mathbf{z}^{(N)} \in \mathcal{K} \triangleq \{\mathbf{y} \in \mathbf{y}^N : \mathbf{y} \in \mathcal{S}^\star\}$ uniformly at random, where $\mathcal{S}^\star$ denotes the ground-truth membership oracle accessible to BoN. Unlike the sequential protocol, batched sampling with an accurate verifier does not guarantee zero sub-optimality, as the algorithm is restricted to selecting from only $N+1$ samples, which may fail to adequately represent the target distribution. Therefore, we begin by deriving a *sufficient condition* on the $N$ to ensure coverage.

**Theorem 3.7** (Maximum admissible batch size of BoN). *Let $\nu_{\mathrm{BoN}}$ denote the sampling distribution induced by BoN with access to the ground-truth membership oracle $\mathcal{S}^\star$. Then $\nu_{\mathrm{BoN}}$ satisfies the coverage constraint, i.e., $\nu_{\mathrm{BoN}} \in \Pi(\beta \mid \mathbf{x})$, only if $N \le \lfloor N_{\max} \rfloor$, where*

$$N_{\max} \triangleq \begin{cases} \infty, & \text{if} \quad \beta \ge (1-s_{r^\star})/s_{r^\star}, \\ \dfrac{\ln\left(1 - \sqrt{(\beta-1)s_{r^\star}(1-s_{r^\star})^{-1}}\right)}{\ln(1-s_{r^\star})}, & \text{if} \quad 1 \le \beta \le (1-s_{r^\star})/s_{r^\star}. \end{cases}$$

We observe that for conservative choices of coverage, BoN admits a maximal batch size. On the other hand, beyond a necessary minimum coverage, the maximum number of samples is an increasing function of $\beta$, and becomes unbounded (as the $\chi^2$-divergence saturates) beyond $\frac{1-s_{r^\star}}{s_{r^\star}}$. Next, we state the sub-optimality of BoN as a function of $N$.

**Theorem 3.8** (Sub-optimality of BoN). *The sub-optimality of the BoN algorithm with access to the ground truth membership oracle $\mathcal{S}^\star$ is*

$$\mathrm{SubOpt}(\mathrm{BoN}) = (1 - \mathrm{s}_{r^\star})^{N+1} - \left(0 \vee 1 - \mathrm{m}_\beta(\mathrm{s}_{r^\star})\right).$$

From Theorem 3.8, as $N$ increases, BoN sub-optimality decreases. However, Theorem 3.7 shows that $N$ cannot grow arbitrarily without inducing constraint violations. For small $\beta$, BoN may even outperform the skyline, whose mass on $\mathcal{S}^\star$ can be strictly less than 1, resulting in negative sub-optimality, albeit at the cost of violating the coverage constraint. More generally, combining Theorems 3.7 and 3.8, we find that for large $\beta$ the batch size $N$ can be chosen freely, yielding vanishing sub-optimality. In contrast, for intermediate $\beta$, restricting $N$ to its maximal admissible value leads to a sub-optimality equal to $1 - \sqrt{(\beta-1)s_{r^\star}/(1-s_{r^\star})} - (0 \vee 1 - m_\beta(s_{r^\star}))$.

### 3.2.2 BATCHED REJECTION SAMPLING (BRS)

Motivated by the infeasibility and constant sub-optimality of BoN in low-coverage regimes, we extend our SRS algorithm to the batched setting, which we call BRS (Algorithm 4). BRS follows the same principles as SRS, with the key distinction that generation is truncated after $N+1$ samples. A batch $\mathbf{Y}^{N+1}$ is drawn in parallel, and rejection sampling is applied to any $N$ of these samples. If none are accepted, the $(N+1)^{\text{th}}$ sample is returned as a fallback. Unlike SRS, however, BRS is *not* a valid transport plan with respect to a target measure defined by a reward $r$, and thus, incurs sub-optimality even when the ground-truth membership oracle is available. Now, we first show that BRS satisfies the coverage constraint for *all* $N$, allowing batch sizes to be chosen freely based on hardware capacity. We then analyze its sub-optimality establishing that it vanishes as $N$ increases.

**Theorem 3.9** (Batch size of BRS). *Let $\nu_{\text{BRS}}$ be the sampling distribution of the BRS algorithm induced by the ground truth membership oracle $\mathcal{S}^\star$. For any prompt $\mathbf{x} \in \mathcal{X}$ and $N \in \mathbb{N}$, we have $\nu_{\text{BRS}} \in \Pi(\beta \mid \mathbf{x})$.*

**Theorem 3.10** (Sub-optimality of BRS). *The sub-optimality of the BRS algorithm with access to the ground truth membership oracle $\mathcal{S}^\star$ is given by*

$$\text{SubOpt}(\text{BRS}) = \text{OTC}(\beta) \left( 1 - \frac{1}{M} \right)^N.$$

We observe that the sub-optimality of BRS decays exponentially in the batch size. Furthermore, setting its envelope to its tightest value $M = \left( \frac{1}{s_{r^\star}} \wedge \frac{m_\beta(s_{r^\star})}{s_{r^\star}} \right)$, we observe that the sub-optimality is $\text{OTC}(\beta)^{N+1} \cdot m_\beta(s_{r^\star})^{-N}$, and it scales exponentially in OTC. This provably shows an improvement in the performance of BRS compared to BoN in the intermediate and low coverage regimes.

## 4 EXPERIMENTAL ANALYSIS

Our empirical study is guided by two central questions:

1. To what extent do the empirical sub-optimality curves align with the three-regimes of theoretical predictions?

2. How sensitive are the algorithms to misspecification of the coverage parameter $s_{\text{ver}}$ used by the algorithms relative to the (unknown) true mass?

We pivot our empirical results on two key performance metrics, sub-optimality and computational complexity. For both metrics, we sweep the coverage budget $\beta$ over a grid spanning the three regimes highlighted by the theory (transport, policy improvement, and saturation). Additionally, for the batched setting, we sweep over the batch size $N+1$. We summarize the key empirical findings in this section, while deferring experimental setup details, construction of ground truth and approximately correct verifiers, and additional results to Appendix P. All curves are averaged over 5,000 episodes, with each algorithm run independently in each episode.

**Sub-optimality.** In sequential protocol, the sub–optimality curves for SRS and SMC in Figure 4 follow the characteristic three–regime geometry predicted by the analysis. In the *small–coverage regime*, sub–optimality increases as $O(\sqrt{\beta})$ and exhibits little policy improvement. *As $\beta$ grows*, the curves bend downward in proportion to the informativeness of the verifier (larger $J$), and finally, plateau at a level determined by $s_{r^\star}$ and $J$. In contrast, AiC aligns with the other methods only under the saturation regime, and otherwise, exhibits constraint violations. The three methods converge in the saturation regimes, achieving the same performance. *Varying the model scale primarily shifts*

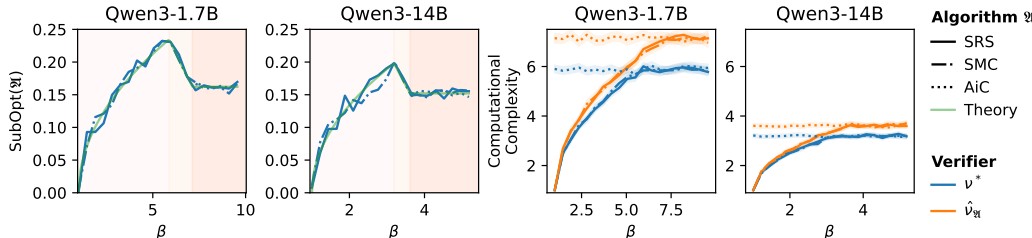

Figure 4: Sub-optimality (left) and computational complexity (right) as functions of $\beta$ for `Qwen3-1.7B` and `Qwen3-14B`. Results are shown for SRS, stochastic SMC, and AiC. Solid green lines denote theoretical predictions as stated in Theorem 3.6. Background shading indicates different coverage regimes, and confidence intervals are shown as shaded bands.

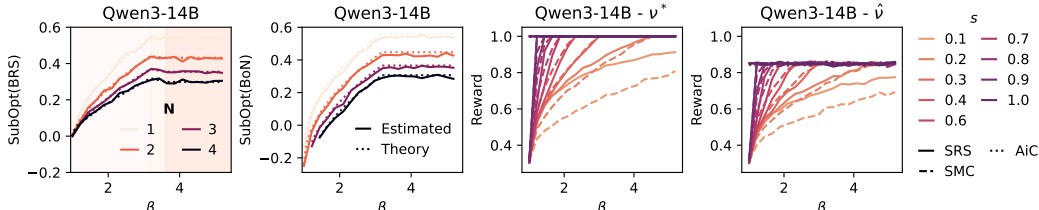

Figure 5: Sub-optimality for `Qwen3-14B` of BRS and BoN with imperfect verifiers (left), and ablation study in $s$ (sensitivity) for SRS, SMC, and AiC (right).

*the saturation level.* Larger Qwen models yield higher $s_{r^\star}$ (stronger base accuracy) and therefore lower residual sub–optimality.

**Computational complexity.** The premise of our experiments in Figure 4 comprises a smaller $s_{\text{ver}}$ compared to $s_{r^\star}$; consequently, we observe that the computational complexity with the approximate verifier (saturating at $\approx 8$ proposals on average for `Qwen3-1.7B`) exceeds the complexity required by the ground truth verifier (saturating at $\approx 6$ proposal on average for `Qwen3-1.7B`). In general, computational complexity for SRS and SMC are identical and scale as $O(\sqrt{\beta})$ before saturating, while AiC has a constant computational complexity as stated in Theorem 3.2.

In the batched protocol, we present a comparison of the sub-optimality of both BRS and BoN under imperfect verifiers in Figure 5 (left), with additional details provided in Appendix O. The sub-optimality is evaluated as a function of $\beta$ across varying batch sizes $N \leq N_{\max}$. Theoretical predictions closely align with empirical results obtained using `Qwen3-14B`. As predicted by Theorems O.1 and O.2, the sub-optimality decreases with increasing $N$ in the presence of imperfect verifiers, reflecting the expected improvement with larger batches.

**Sensitivity to $s_{\text{ver}}$.** Since $s_{\text{ver}}$ is typically unavailable in real-world settings, we conduct an ablation study by setting $s_{\text{ver}} = s$ for various choices of $s$. The two rightmost plots in Figure 5 show the reward obtained by the algorithms when a value $s$ selected from the set shown in the legend is used instead of the one induced by the verifier under examination. Each curve corresponds to a different assumed value of $s$, and illustrates how mismatched assumptions about verifier accuracy affect the reward. Interestingly, when $s = 1$, all three algorithms reduce to the AiC algorithm. This is because (i) RS envelope becomes $M = 1$. Thus, the first check in SMC becomes identical to the SRS acceptance condition. (ii) The Radon-Nikodym derivative function becomes $\eta_r(\mathbf{y}) = \mathbb{1}\{\mathbf{y} \in \mathcal{S}_r\}$. Thus, for $s = 1$, all methods restrict support to $\mathcal{S}_r$, and behave identically, as reflected in the overlapping curves at that point. Also, an interesting pattern emerges when comparing SRS and SMC across different assumed values of $s$. Specifically, the two methods exhibit matching performance when $s$ is aligned with the true verifier accuracy, i.e., at $s = 0.31$ in ground truth case, and $s = 0.27$ when using the approximate verifier. Notably, SMC underperforms relative to SRS when the assumed $s$ is smaller than the true value ($s \leq 0.31$ or $s \leq 0.27$), and outperforms SRS when the assumed $s$ is greater ($s \geq 0.31$ or $s \geq 0.27$). This crossover behavior illustrates the sensitivity of SMC to over- or under-estimating verifier accuracy, and highlights that SMC may be advantageous in high-$s$ regimes, whereas SRS is more robust when verifier confidence is low.

## 5 CONCLUSION AND FUTURE WORKS

We cast test-time verification through the lens of optimal transport. By positing it as a sampling problem, we analyzed how generator's coverage, verifier's accuracy, and sampling algorithms jointly determine sub-optimality and computational complexity. Our analysis, supported by empirical evidence, reveals a three-regime structure in the sub-optimality–coverage tradeoff: a *transport regime*, where sub-optimality is dominated by transport cost; a *policy-improvement regime*, where sampling can counteract transport cost depending on the verifier's ROC; and a *saturation regime*, where sub-optimality plateaus at a level dictated by the verifier's Youden's index. These dynamics are exhibited by both the sequential and batched algorithms studied. Notably, *rejection sampling–type methods are advantageous under low coverage, while best-of-$N$ approaches excel under liberal coverage.*

**Future Works.** Our study also raises several open questions. Analytically, extending from ratio-based to difference-based coverage remains unexplored. More broadly, moving beyond verifiable rewards toward general reward models for inference-time alignment is an important next step. Finally, our premise highlights a fundamental open problem in sampling: *how can we sample from a target distribution given only proposals, when the target-to-proposal likelihood ratio is partially or fully unknown and must be estimated from samples?* We conjecture that any such algorithm must explicitly balance exploration, i.e., estimating the likelihood ratio with sufficient confidence, against exploitation, i.e., using the estimate to make acceptance or stopping decisions.

### ACKNOWLEDGMENTS

This work received funding from the SNS JU project 6G-GOALS under the EU's Horizon program Grant Agreement No. 101139232; and by the INFORMED-AI Hub under Grant EP/Y028732/1. D. Basu acknowledges the ANR JCJC project REPUBLIC (ANR-22-CE23-0003-01), PEPR project FOUNDRY (ANR23-PEIA-0003), and Inria-ISI Kolkata associate team SeRAI for support.

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

# Appendix

## Table of Contents

# A  LITERATURE REVIEW

A crucial bottleneck in pre- and post-training pipelines for large language models (LLMs) is the dwindling supply of high-quality training data, constrained by privacy, security, and cost concerns (Liu et al., 2024; Villalobos et al., 2024). This trend threatens saturation along the train-time scaling axis. In response to such bottlenecks in scaling laws, OpenAI introduced an alternate axis – *test-time scaling* – and showcased its potential through the OpenAI-o1 release. This shift has delivered substantial gains across diverse benchmarks (Jaech et al., 2024). Ever since, the community has witnessed a plethora of investigations into attributes including, but not limited to, scaling laws, methodologies, trade-offs, and a theoretically-grounded understanding of the new scaling axis. Test-time scaling is mostly realized through two approaches- *verifier-free* and *verifier-based* (Setlur et al., 2025).

**Verifier-free vs. verifier-based methods. Verifier-free methods** involve performing supervised fine-tuning (SFT) on pre-trained LLMs with *expert traces*, i.e., high-quality step-by-step rationales that directly supervise the reasoning process. Expert traces can come from diverse sources, such as human-written or curated solutions (e.g., GSM8K (Cobbe et al., 2021)), distilled chain-of-thought (CoT) from stronger teacher models (Muennighoff et al., 2025), reasoning trajectories obtained via search procedures (Gandhi et al., 2024; Moon et al., 2024), self-bootstrapped rationales where only correct generations are retained (Zelikman et al., 2022), and rationale distillation techniques for transferring reasoning ability across models (Hsieh et al., 2023). On the other hand, **verifier-based methods** deploy a *verifier*, a reward-signal apparatus, for guiding the response generation. Verifier assigns a binary $(0 / 1)$ value assessing the generation quality, especially in the objective tasks such as math and coding. A verifier is construed from a domain-specific *de facto* ground truth, such as constructing unit tests for coding and correct solutions for mathematical tasks. Verification has been leveraged during both training (also known as reinforcement learning with verifiable rewards) (Guo et al., 2025; Luo et al., 2025; Team et al., 2025), and inference (Cobbe et al., 2021). This approach has exhibited strong test-time scaling performance. Indeed, Setlur et al. (2025) show that verifier-based methods provably outperform verifier-free methods in test-time scaling.

**Sequential vs. parallel compute.** A complementary concern for test-time scaling methods is how they spend their test-time compute budget. The *reasoning* models spend their entire budget sequentially by refining a single trajectory over multiple steps to curate a longer and more accurate response. The sequential sampling process may be verifier-based, e.g., through process reward models (Liao et al., 2025), or verifier-free (Chen et al., 2023). On the other hand, *resampling* methods adopt a parallel compute mode by dividing its budget to generate multiple responses, and then, distilling a winning response. Popular resampling methods are verifier-based, leveraging a verifier (e.g., unit tests, reward models trained on ground truth responses for objective tasks, etc.) to distill a winning response from its generations (Huang et al., 2025a; Beirami et al., 2024; Cobbe et al., 2021). The focus of this investigation is on **verifier-based resampling** methods. We measure computational complexity via the expected number of proposals, which is a hardware-agnostic quantity that arises naturally in our analysis. In practice, however, wall-clock latency also depends on GPU parallelism, model size, and verifier overhead. Batched methods can exploit accelerator parallelism to evaluate many candidates simultaneously and may therefore achieve lower latency than sequential methods, even when they use more total proposals. Conversely, sequential methods can continue sampling until a high-quality solution is found and typically achieve better sub-optimality for a given proposal budget, highlighting a practical tradeoff between solution quality and runtime.

**Best-of-$N$ (BoN) sampling.** The most popular verifier-based test-time scaling method is BoN sampling (Brown et al., 2024). BoN generates $N$ independent responses per prompt, and chooses a winner randomly from the set of correct responses deemed by a verifier. Assuming access to an accurate verifier, Brown et al. (2024) analyze the $pass@N$ metric, i.e., the average fraction of prompts with at least one correct response, and observe an approximate *power-law scaling* with $N$. Extending this analysis, Schaeffer et al. (2025) establish an *exponential* per-instance scaling law attributing the aggregate power law to the heavy-tailed distribution over prompts. On a complementary note, Beirami et al. (2024); Mroueh (2024) analyze the deviation of the BoN alignment policy from the reference policy by computing tight upper-bounds on their KL-divergence. Yet another characteristic, the *sample complexity* of BoN to generate correct responses for objective tasks is investigated by Huang et al. (2025b). While such scaling laws and divergence bounds are informative, they do not directly address the central goal of resampling: *how well can we approximate the verifier-induced*

*optimal policy that maximizes expected reward?* Variations of BoN, addressing aspects such as $N$-estimation and adaptivity, and finding alternate scoring mechanisms have also been explored. For instance, Wang et al. (2025) truncates BoN generations based on an estimation budget formed by solving an optimization problem, showcasing computational improvement over BoN. Huang et al. (2025c) preaches why confidence score is preferable compared to reward scores, and thresholds it for "confident" test-time scaling. However, the choice of threshold is ad hoc.

**Approximately correct verifiers.** Most of the studies on BoN assume access to an accurate verifier that rarely holds in practice. For example, unit tests can miss the edge cases, and verifiers for math benchmarks often capture only a *subset* of valid responses. A relevant case is the default GSM8K evaluation in `lm-evaluation-harness` (Gao et al., 2024), which extracts the first match to the pattern "`The answer is (-?[0-9.,]+)`"; many correct generations that deviate from this template are thus marked incorrect. These limitations underscore the need to explicitly account for verifier imperfections in the design and analysis of test-time scaling methods — a dimension largely absent from the literature. We are aware of two investigations accounting for verifier (aka reward model) imperfections in sampling algorithms. Aminian et al. (2025) analyze BoN under a per-prompt mean squared error (MSE) constraint on the verifier. Huang et al. (2025a) adopt the same framework to show that BoN's average reward may fail to scale with $N$ under limited coverage, motivating a rejection sampling (RS) variant that alleviates this issue. Concurrently, Dorner et al. (2025) study test-time scaling with approximately correct verifiable rewards, characterizing how a verifier's region of convergence (ROC) mediates the trade-off between accuracy and compute. Our investigation complements these perspectives by focusing on generator coverage and showing how coverage, together with the ROC, determines the *exact* sub-optimality of a sampling method, rather than only its asymptotic accuracy-compute profile. Here, sub-optimality is defined as the difference between the average reward obtained from the verifier while using an optimal policy and that of the sampling algorithm.

# B ALGORITHM PSEUDO-CODES

---

**Algorithm 1:** Accept-if-Correct (AiC)

---

**Input:** Prompt $\mathbf{x} \in \mathcal{X}$, generator $\pi_{\text{ref}}(\cdot \mid \mathbf{x})$, verifier set $\widehat{\mathcal{S}}(\mathbf{x})$

**for** $n = 1, 2, \dots$ **do**

    Sample $\mathbf{Y}_n \sim \pi_{\text{ref}}(\cdot \mid \mathbf{x})$;

    **if** $\mathbf{Y}_n \in \widehat{\mathcal{S}}(\mathbf{x})$ **then**

        **return** $\mathbf{Y}_n$ `// accept if correct`

    **else**

        continue `// resample`

---

**Algorithm 2:** Sequential Rejection Sampling (SRS)

---

**Input:** Prompt $\mathbf{x} \in \mathcal{X}$; generator $\pi_{\text{ref}}(\cdot \mid \mathbf{x})$; verifier set $\widehat{\mathcal{S}}(\mathbf{x})$; envelope $M$; $\widehat{s} \triangleq \mu(\widehat{\mathcal{S}}(\mathbf{x}))$; constraint $\beta$

**for** $n = 1, 2, \dots$ **do**

    Sample $\mathbf{Y}_n \sim \pi_{\text{ref}}(\cdot \mid \mathbf{x})$ and $u \sim \text{Unif}[0, 1]$;

    Compute $\widehat{\eta}(\mathbf{Y}_n)$ by plugging $\widehat{s}$ in (2);

    **if** $\mathbf{Y}_n \in \widehat{\mathcal{S}}(\mathbf{x})$ **then**

        **return** $\mathbf{Y}_n$ `// accept if verified correct`

    **else if** $\frac{1}{M} \widehat{\eta}(\mathbf{Y}_n) \geq u$ **then**

        **return** $\mathbf{Y}_n$ `// accept (incorrect) via RS to satisfy coverage`

    **else**

        continue `// reject and resample`

---

**Algorithm 3:** Sequential Maximal Coupling (SMC)

---

**Input:** Prompt $\mathbf{x} \in \mathcal{X}$; generator $\pi_{\text{ref}}(\cdot \mid \mathbf{x})$; verifier set $\widehat{\mathcal{S}}(\mathbf{x})$; $\widehat{s} \triangleq \mu(\widehat{\mathcal{S}}(\mathbf{x}))$; constraint $\beta$

**for** $n = 1, 2, \dots$ **do**

    Sample $\mathbf{Y}_n \sim \pi_{\text{ref}}(\cdot \mid \mathbf{x})$ and $u \sim \text{Unif}[0, 1]$;

    Compute $\widehat{\eta}(\mathbf{Y}_n)$ by plugging $\widehat{s}$ in (2);

    **if** $\widehat{\eta}(\mathbf{Y}_n) \geq u$ **then**

        **return** $\mathbf{Y}_n$ `// accept if coverage is satisfied`

    **else**

        `// do not advance n:  keep drawing until verified-correct`

        **while** $\mathbf{Y}_n \notin \widehat{\mathcal{S}}(\mathbf{x})$ **do**

            Sample $\mathbf{Y}_n \sim \pi_{\text{ref}}(\cdot \mid \mathbf{x})$;

        **return** $\mathbf{Y}_n$ `// first verified-correct draw`

---

**Algorithm 4:** Batched Rejection Sampling (BRS)

---

**Input:** Prompt $\mathbf{x} \in \mathcal{X}$; generator $\pi_{\text{ref}}(\cdot \mid \mathbf{x})$; verifier set $\widehat{\mathcal{S}}(\mathbf{x})$; envelope $M$; $\widehat{s} \triangleq \mu(\widehat{\mathcal{S}}(\mathbf{x}))$; batch size $N + 1$; constraint $\beta$

Sample $\mathbf{Y}^{N+1} \triangleq (\mathbf{Y}_1, \dots, \mathbf{Y}_{N+1})$ i.i.d. from $\pi_{\text{ref}}(\cdot \mid \mathbf{x})$;

Draw $u_1, \dots, u_{N+1} \sim \text{Unif}[0, 1]$;

**for** $i = 1, \dots, N$ **do**

    Compute $\widehat{\eta}(\mathbf{Y}_i)$ by plugging $\widehat{s}$ in (2);

    **if** $\mathbf{Y}_i \in \widehat{\mathcal{S}}(\mathbf{x})$ **then**

        **return** $\mathbf{Y}_i$ `// accept if verified correct`

    **else if** $\frac{1}{M} \widehat{\eta}(\mathbf{Y}_i) \geq u_i$ **then**

        **return** $\mathbf{Y}_i$ `// accept (incorrect) via RS to satisfy coverage`

**return** $\mathbf{Y}_{N+1}$ `// return the last sample if none accepted`

---

## C  TARGET-TO-PROPOSAL RADON-NIKODYM DERIVATIVE (PROOF OF THEOREM 2.1)

Finding the optimal policy $\nu_r$ is equivalently solving the following constrained optimization problem.

$$\mathcal{P}(\beta) \quad \triangleq \quad \max_{\eta \geq 0} \int_{\mathcal{S}_r} \eta \, \mathrm{d}\mu \quad \text{s.t.} \quad \int \eta^2 \, \mathrm{d}\mu \leq \beta \,, \quad \text{and} \quad \int \eta \, \mathrm{d}\mu = 1 \,.$$

Let us denote the value of $\mathcal{P}(\beta)$ by $m$, i.e., $m \triangleq \int_{\mathcal{S}_r} \eta_r \, \mathrm{d}\mu$. Using Cauchy-Schwarz inequality, we have

$$\left( \int_{\mathcal{S}_r} \eta_r \, \mathrm{d}\mu \right)^2 \leq \int_{\mathcal{S}_r} \eta_r^2 \, \mathrm{d}\mu \cdot \int_{\mathcal{S}_r} \mathrm{d}\mu \,,$$

which yields:

$$\int_{\mathcal{S}_r} \eta_r^2 \, \mathrm{d}\mu \geq \frac{1}{s_r} m^2 \,. \tag{4}$$

Similarly, from the fact that

$$\left( \int_{\overline{\mathcal{S}}_r} \eta_r \, \mathrm{d}\mu \right)^2 \leq \int_{\overline{\mathcal{S}}_r} \eta_r^2 \, \mathrm{d}\mu \cdot \int_{\overline{\mathcal{S}}_r} \mathrm{d}\mu \,,$$

we obtain:

$$\int_{\overline{\mathcal{S}}_r} \eta_r^2 \, \mathrm{d}\mu \geq \frac{1}{1 - s_r} (1 - m)^2 \,. \tag{5}$$

Combining (4) and (5), we have

$$\beta \geq \int_{\mathcal{Y}} \eta^2 \, \mathrm{d}\mu \geq \frac{1}{s_r} m^2 + \frac{1}{1 - s_r} (1 - m)^2 \,. \tag{6}$$

Rearranging (6), we obtain:

$$m \leq \sqrt{s_r (1 - s_r)(\beta - 1)} + s \,.$$

Furthermore, since $m$ is the mass that the optimal measure puts on the set of correct responses $\mathcal{S}_r$, we have

$$m \leq \left( 1 \wedge \sqrt{s_r (1 - s_r)(\beta - 1)} + s_r \right) \,. \tag{7}$$

Next, we will show that the upper-bound on $m$ is tight. Specifically, we will construct a valid $\eta_r$ such that (7) holds with equality, noting that Cauchy-Schwarz is tight for constant functions. For some $p_r \in \mathbb{R}$ and $q_r \in \mathbb{R}$, let us set

$$\eta_r(\mathbf{y}) = \begin{cases} p_r, & \mathbf{y} \in \mathcal{S} \,, \\ q_r, & \mathbf{y} \notin \mathcal{S} \,. \end{cases}$$

Since $\nu_r$ is a probability measure, we have

$$1 = \int \eta_r \, \mathrm{d}\mu = \int_{\mathcal{S}_r} \eta_r \, \mathrm{d}\mu + \int_{\overline{\mathcal{S}}_r} \eta_r \, \mathrm{d}\mu = \underbrace{p_r \cdot s_r}_{= m} + q_r \cdot (1 - s_r) \,. \tag{8}$$

From (8) we have

$$p_r = \frac{m}{s_r} \,, \quad \text{and} \quad q_r = \frac{1 - m}{1 - s_r} \,.$$

The proof concludes by setting $m = (1 \wedge \sqrt{s_r (1 - s_r)(\beta - 1)} + s_r)$.

# D AUXILIARY LEMMAS

## D.1 OPTIMAL TRANSPORT COST (PROOF OF LEMMA 3.1)

Let us define the total variation (TV) distance between measures $\mu$ and $\nu$ defined on a common measurable space $(\mathcal{Y}, \mathfrak{B}(\mathcal{Y}))$ as

$$D_{\mathsf{TV}}(\mu \| \nu) \triangleq \frac{1}{2} \cdot \int_{\mathcal{Y}} \left| \mu(\mathrm{d}\mathbf{y}) - \nu(\mathrm{d}\mathbf{y}) \right| . \tag{9}$$

We have

$$
\begin{aligned}
\mathrm{OTC}(\beta) &= \min_{\rho \in \mathcal{M}(\mu, \nu^\star)} \int \mathbb{1}\{\mathbf{y} \neq \mathbf{z}\} \, \mathrm{d}\rho(\mathbf{y}, \mathbf{z}) \\
&= \min_{\rho \in \mathcal{M}(\mu, \nu^\star)} \mathbb{P}_{\mathbf{y}, \mathbf{z} \sim \rho}(\mathbf{y} \neq \mathbf{z}) \\
&= D_{\mathsf{TV}}(\mu \| \nu^\star) \tag{10} \\
&\stackrel{(9)}{=} \frac{1}{2} \left( \int_{\mathcal{S}^\star} |\mu(\mathrm{d}\mathbf{y}) - \nu^\star(\mathrm{d}\mathbf{y})| + \int_{\overline{\mathcal{S}}^\star} |\mu(\mathrm{d}\mathbf{y}) - \nu^\star(\mathrm{d}\mathbf{y})| \right) \\
&\stackrel{(2)}{=} \frac{1}{2} \left( \left| \left(1 \wedge m_\beta(s_{r^\star})\right) - s_{r^\star} \right| + \left| \left(0 \vee 1 - m_\beta(s_{r^\star})\right) - (1 - s_{r^\star}) \right| \right) \\
&= \left(1 \wedge m_\beta(s_{r^\star})\right) - s_{r^\star} , \tag{11}
\end{aligned}
$$

where (10) is a well known result, see, for example, (Villani et al., 2008, page 22), and (11) follows by noting that $m_\beta(s_{r^\star}) \geq s_{r^\star}$ by definition, since $m_\beta(s_{r^\star})$ is the mass that the *optimal policy* puts on $\mathcal{S}^\star$, and must be at least equal to $s_{r^\star}$.

## D.2 BON SAMPLING DISTRIBUTION

In this subsection, we characterize the BoN sampling distribution, where the BoN algorithm has access to a membership oracle $\mathcal{S}$, such that $r(\mathbf{x}, \mathbf{y}) = \mathbb{1}\{\mathbf{y} \in \mathcal{S}\}$ for any prompt $\mathbf{x} \in \mathcal{X}$ and response $\mathbf{y} \in \mathcal{Y}$. Note that the analysis for BoN sampling distribution presented in (Beirami et al., 2024) **does not apply** to our setting, since it assumes a *strictly monotonic* reward function. We have the following result. Let us denote $s \triangleq \int_{\mathcal{S}} r \, \mathrm{d}\mu$.

**Lemma D.1** (BoN – Radon-Nikodym derivative). *Let $\nu_{\mathrm{BoN}}^{(N)}$ denote the sampling distribution induced by BoN with batch size $N + 1$ and access to a membership oracle $\mathcal{S}$. We have*

$$\frac{\mathrm{d}\nu_{\mathrm{BoN}}^{(N)}}{\mathrm{d}\mu}(\mathbf{y}) \triangleq \begin{cases} (1 - s)^N, & \text{if} \quad \mathbf{y} \notin \mathcal{S}, \\ \frac{1}{s}\left(1 - (1 - s)^{N+1}\right), & \text{if} \quad \mathbf{y} \in \mathcal{S} . \end{cases}$$

*Proof.* For any set $\mathcal{A} \in \mathcal{Y}$, we have

$$
\begin{aligned}
\nu_{\mathrm{BoN}}^{(N)}(\mathcal{A}) &= \sum_{n \in [N+1]} \mathbb{P}\left(\mathbf{Y}_n \in \mathcal{A}, \text{ response } n \text{ is selected}\right) \\
&= (N + 1)\mathbb{P}\left(\mathbf{Y}_n \in \mathcal{A}, \text{ response } n \text{ is selected}\right) ,
\end{aligned}
$$

which follows from the independence of the sampled response $(\mathbf{Y}_1, \cdots, \mathbf{Y}_{N+1})$. Next, note that

$$\mathbb{P}\left(\mathbf{Y}_1 \in \mathcal{A}, \text{ response } 1 \text{ is selected}\right)$$

$$= \int \mathbb{P}\left(\mathbf{Y}_1 \in \mathcal{A}, \text{ response } 1 \text{ is selected} \mid \mathbf{Y}_1 = \mathbf{y}\right) \mu(\mathrm{d}\mathbf{y})$$

$$= \int \mathbb{P}\left(\mathbf{Y}_1 \in \mathcal{A} \mid \text{ response } 1 \text{ is selected}, \mathbf{Y}_1 = \mathbf{y}\right) \cdot \mathbb{P}\left(\text{response } 1 \text{ is selected} \mid \mathbf{Y}_1 = \mathbf{y}\right) \mu(\mathrm{d}\mathbf{y})$$

$$= \int \mathbb{P}\left(\mathbf{Y}_1 \in \mathcal{A} \mid \mathbf{Y}_1 = \mathbf{y}\right) \cdot \mathbb{P}\left(\text{response } 1 \text{ is selected} \mid \mathbf{Y}_! = \mathbf{y}\right) \mu(\mathrm{d}\mathbf{y})$$

$$= \int \mathbb{1}\{\mathbf{y} \in \mathcal{A}\} \cdot \mathbb{P}(\text{response 1 is selected} \mid \mathbf{Y}_1 = \mathbf{y})\mu(\mathrm{d}\mathbf{y}) ,$$

which implies that

$$\nu_{\mathrm{BoN}}^{(N)}(\mathcal{A}) = (N+1) \int \mathbb{1}\{\mathbf{y} \in \mathcal{A}\} \cdot \mathbb{P}(\text{response 1 is selected} \mid \mathbf{Y}_1 = \mathbf{y})\mu(\mathrm{d}\mathbf{y}) .$$

Thus, the Radon-Nikodym derivative of $\nu_{\mathrm{BoN}}^{(N)}$ with respect to the proposal $\mu$ is given by

$$\frac{\mathrm{d}\nu_{\mathrm{BoN}}^{(N)}}{\mathrm{d}\mu}(\mathbf{y}) = (N+1) \cdot \mathbb{P}(\text{response 1 is selected} \mid \mathbf{Y}_1 = \mathbf{y})$$

$$= (N+1) \underbrace{\mathbb{P}(\text{response 1 is selected}, \ \mathbf{Y}_1 \notin \mathcal{S} \mid \mathbf{Y}_1 = \mathbf{y})}_{\triangleq T_1}$$

$$+ (N+1) \cdot \underbrace{\mathbb{P}(\text{response 1 is selected}, \ \mathbf{Y}_1 \in \mathcal{S} \mid \mathbf{Y}_1 = \mathbf{y})}_{\triangleq T_2} .$$

Expanding $T_1$, we have

$$T_1 = \mathbb{P}(\mathbf{Y}_j \notin \mathcal{S} \ \forall \ j \in [N+1], \ \text{response 1 is selected} \mid \mathbf{Y}_1 = \mathbf{y})$$

$$= \mathbb{P}(\text{response 1 is selected} \mid \mathbf{Y}_1 = \mathbf{y}, \ \mathbf{Y}_j \notin \mathcal{S} \ \forall \ j \in [N+1])$$

$$\times \mathbb{P}(\mathbf{Y}_j \notin \mathcal{S} \ \forall \ j \in [N+1] \mid \mathbf{Y}_1 = \mathbf{y})$$

$$= \frac{1}{N+1} \prod_{j \in [N+1]} \mathbb{P}(\mathbf{Y}_j \notin \mathcal{S} \mid \mathbf{Y}_1 = \mathbf{y})$$

$$= \frac{1}{N+1} \cdot (1-s)^N \mathbb{1}\{\mathbf{y} \notin \mathcal{S}\} . \tag{12}$$

Furthermore, we have

$$T_2 = \sum_{m \in [N+1]} \mathbb{P}\Big((\mathbf{Y}_1, \cdots, \mathbf{Y}_m) \in \mathcal{S}^{\otimes m}, \ (\mathbf{Y}_{m+1}, \cdots, \mathbf{Y}_{N+1}) \notin \mathcal{S}^{\otimes N-m},$$

$$\text{response 1 is selected} \mid \mathbf{Y}_1 = \mathbf{y}\Big)$$

$$= \sum_{m=0}^{N+1} \frac{\binom{N}{m-1}}{m} \cdot s^{m-1} \cdot (1-s)^{N-m+1} \mathbb{1}\{\mathbf{y} \in \mathcal{S}\}$$

$$= \sum_{m=0}^{N} \frac{\binom{N}{m}}{m+1} \cdot s^m \cdot (1-s)^{N-m} \mathbb{1}\{\mathbf{y} \in \mathcal{S}\} . \tag{13}$$

Combining (12) and (13), we get:

$$\frac{\mathrm{d}\nu_{\mathrm{BoN}}^{(N)}}{\mathrm{d}\mu}(\mathbf{y}) \triangleq \begin{cases} (1-s)^N, & \text{if} \quad \mathbf{y} \notin \mathcal{S}, \\ (N+1) \cdot \sum_{m=0}^{N} \frac{\binom{N}{m}}{m+1} \cdot s^m \cdot (1-s)^{N-m}, & \text{if} \quad \mathbf{y} \in \mathcal{S} . \end{cases} \tag{14}$$

Furthermore, note that $\int_0^1 t^m \, \mathrm{d}t = \frac{1}{m+1}$. Hence, we can further simplify (14) as follows.

$$\cdot \sum_{m=0}^{N} \frac{\binom{N}{m}}{m+1} \cdot s^m \cdot (1-s)^{N-m} = \cdot \sum_{m=0}^{N} \binom{N}{m} \cdot s^m \cdot (1-s)^{N-m} \int_0^1 t^m \, \mathrm{d}t$$

$$= \int_0^1 \sum_{m=0}^{N} \binom{N}{m} (st)^m \cdot (1-s)^{N-m} \, \mathrm{d}t$$

$$= \int_0^1 (1-s+st)^N \, \mathrm{d}t$$

$$= \frac{1}{s} \cdot \frac{1-(1-s)^{N+1}}{N+1} ,$$

which yields the desired result. ∎

## D.3 BRS SAMPLING DISTRIBUTION

In this subsection, we characterize the BRS sampling distribution, where we assume BRS' acces to a membership oracle $\mathcal{S}$ obtainable through a verifier $r(\mathbf{x}, \mathbf{y}) = \mathbb{1}\{\mathbf{y} \in \mathcal{S}\}$ for any prompt $\mathbf{x} \in \mathcal{X}$ and response $\mathbf{y} \in \mathcal{Y}$. Denoting $s \triangleq \int_{\mathcal{S}} r \, \mathrm{d}\mu$, we have the following lemma.

**Lemma D.2** (BRS – Radon-Nikodym derivative). *Let $\nu_{\mathrm{BRS}}^{(N)}$ denote the sampling distribution induced by BRS with batch size $N + 1$ and access to a membership oracle $\mathcal{S}$. Furthermore, let $\nu$ denote the optimal policy in $\Pi(\beta \mid \mathbf{x})$ induced by $\mathcal{S}$, i.e., $\nu \triangleq \arg\max_{\rho \in \Pi(\beta \mid \mathbf{x})} \int_{\mathcal{S}} \mathrm{d}\rho$. We have*

$$\frac{\mathrm{d}\nu_{\mathrm{BRS}}^{(N)}}{\mathrm{d}\mu}(\mathbf{y}) = \left(1 - \left(1 - \frac{1}{M}\right)^N\right) \frac{\mathrm{d}\nu}{\mathrm{d}\mu}(\mathbf{y}) + \left(1 - \frac{1}{M}\right)^N .$$

*Proof.* Recall that BRS obtains a batch of $N + 1$ samples, which we denote by $\mathbf{y}^{N+1} \triangleq (\mathbf{y}_1, \cdots, \mathbf{Y}_{N+1})$. Denote the target-to-proposal Radon-Nikodym derivative that BRS uses to accept sample $\mathbf{y}$ by $\eta(\mathbf{y})$. For our analysis, this corresponds to (2) induced by $\mathcal{S}$. The conditional probability kernel for the BRS sampling strategy, which we denote by $K(\mathbf{y}^{N+1}, \mathrm{d}\mathbf{z})$, is given by

$$K(\mathbf{y}^{N+1}, \mathrm{d}\mathbf{z}) = \sum_{n \in [N]} \left(\frac{1}{M} \eta(\mathbf{y}_n) \prod_{j < n} \left(1 - \frac{1}{M} \eta(\mathbf{y}_j)\right)\right) \cdot \delta_{\mathbf{y}_n}(\mathrm{d}\mathbf{z})$$
$$+ \left(\prod_{n \in \mathbb{N}} \left(1 - \frac{1}{M} \eta(\mathbf{y}_n)\right)\right) \mu(\mathrm{d}\mathbf{z}) .$$

The BRS coupling is then obtained as

$$\rho_{\mathrm{BRS}}^{(N)} = K(\mathbf{y}^{N+1}, \mathrm{d}\mathbf{z}) \cdot \mu^{\otimes(N+1)}(\mathrm{d}\mathbf{y}^{N+1}) . \tag{15}$$

Marginalizing (15) with respect to $\mathbf{y}^{N+1}$, we obtain

$$\nu_{\mathrm{BRS}}^{(N)}(\mathrm{d}\mathbf{z}) = \int \rho(\mathrm{d}\mathbf{y}^{N+1}, \mathrm{d}\mathbf{z})$$

$$= \int \sum_{n \in [N]} \left(\frac{1}{M} \eta(\mathbf{y}_n) \prod_{j < n} \left(1 - \frac{1}{M} \eta(\mathbf{y}_j)\right)\right) \cdot \delta_{\mathbf{y}_n}(\mathrm{d}\mathbf{z}) \mu^{\otimes(N+1)}(\mathrm{d}\mathbf{y}^{N+1})$$
$$+ \int \left(\prod_{n \in \mathbb{N}} \left(1 - \frac{1}{M} \eta(\mathbf{y}_n)\right)\right) \mu^{\otimes(N+1)}(\mathrm{d}\mathbf{y}^{N+1}) \mu(\mathrm{d}\mathbf{z})$$

$$= \sum_{n \in [N]} \left(\prod_{j < n} \int \left(1 - \frac{1}{M} \frac{\mathrm{d}\nu}{\mathrm{d}\mu}(\mathbf{y}_j) \mu(\mathrm{d}\mathbf{y}_j)\right)\right) \cdot \left(\int \frac{1}{M} \mathrm{d}\nu(\mathbf{y}_n) \delta_{\mathbf{y}_n}(\mathrm{d}\mathbf{z})\right)$$
$$+ \mu(\mathrm{d}\mathbf{z}) \prod_{n \in [N]} \int \left(1 - \frac{1}{M} \frac{\mathrm{d}\nu}{\mathrm{d}\mu}(\mathbf{y}_n)\right) \mu(\mathrm{d}\mathbf{y}_n)$$

$$= \frac{1}{M} \nu(\mathrm{d}\mathbf{z}) \cdot \sum_{n \in [N]} \left(1 - \frac{1}{M}\right)^{n-1} + \left(1 - \frac{1}{M}\right)^N \mu(\mathrm{d}\mathbf{z})$$

$$= \left(1 - \left(1 - \frac{1}{M}\right)^N\right) \nu(\mathrm{d}\mathbf{z}) + \left(1 - \frac{1}{M}\right)^N \mu(\mathrm{d}\mathbf{z}) ,$$

and the lemma readily follows. ∎

## E PROPERTIES OF ROC

In this section, we briefly review the properties of the ROC for completeness and introduce useful definitions that will be leveraged in our subsequent analysis. Recall that $s_{\widehat{r}} = \mu(\widehat{\mathcal{S}})$.

- *True positives (TP):* samples correctly identified by the verifier, i.e., $\widehat{\mathcal{S}} \cap \mathcal{S}^\star$. The reference mass is $\mathrm{TP} \triangleq \mu(\widehat{\mathcal{S}} \cap \mathcal{S}^\star)$. *False positives (FP):* incorrect responses accepted as correct, i.e., $\widehat{\mathcal{S}} \setminus \mathcal{S}^\star$, with mass $\mathrm{FP} \triangleq \mu(\widehat{\mathcal{S}} \setminus \mathcal{S}^\star)$.

- *False negatives (FN):* correct responses rejected by the verifier, i.e., $\mathcal{S}^\star \setminus \widehat{\mathcal{S}}$, with mass $\mathrm{FN} \triangleq \mu(\mathcal{S}^\star \setminus \widehat{\mathcal{S}})$. *True negatives (TN):* incorrect responses correctly rejected, i.e., $\mathcal{Y} \setminus (\mathcal{S}^\star \cup \widehat{\mathcal{S}})$, with mass $\mathrm{TN} \triangleq \mu(\mathcal{Y} \setminus (\mathcal{S}^\star \cup \widehat{\mathcal{S}}))$.

- *True positive rate (TPR):* the fraction of true positives among all ground-truth correct responses:
$$\mathrm{TPR} \;=\; \frac{\mathrm{TP}}{\mathrm{TP} + \mathrm{FN}} \;=\; \frac{1}{s_{r^\star}} \mu(\mathcal{S}^\star \cap \widehat{\mathcal{S}}).$$
Thus, $\mathrm{TP} = s_{r^\star} \cdot \mathrm{TPR}$.

- *False positive rate (FPR):* the fraction of false positives among all ground-truth incorrect responses:
$$\mathrm{FPR} \;=\; \frac{\mathrm{FP}}{\mathrm{FP} + \mathrm{TN}} \;=\; \frac{1}{1 - s_{r^\star}} \mu(\widehat{\mathcal{S}} \setminus \mathcal{S}^\star).$$
Thus, $\mathrm{FP} = (1 - s_{r^\star}) \cdot \mathrm{FPR}$.

- It follows that
$$s_{\widehat{r}} \;=\; \mu(\mathcal{S}^\star \cap \widehat{\mathcal{S}}) + \mu(\widehat{\mathcal{S}} \setminus \mathcal{S}^\star) \;=\; s_{r^\star} \cdot \mathrm{TPR} + (1 - s_{r^\star}) \cdot \mathrm{FPR} = s_{\mathrm{ver}}. \tag{16}$$

- Likewise,
$$\mathrm{FN} \;=\; s_{r^\star} \cdot (1 - \mathrm{TPR}). \tag{17}$$

## F  AiC Properties (Proof of Theorem 3.2)

**Computational complexity.**  For AiC, the acceptance probability is given by

$$p_{\mathrm{AiC}} \;\triangleq\; \mathbb{P}(\mathbf{Z} = \mathbf{Y} \mid \mathbf{Y} \sim \pi_{\mathrm{ref}}(\cdot \mid \mathbf{X})) \;=\; \int_{\widehat{\mathcal{S}}} \mu(\mathrm{d}\mathbf{y}) \;=\; s_{\widehat{r}} \,.$$

As shown in Appendix E, since $s_{\widehat{r}} = s_{\mathrm{ver}}$, we have $p_{\mathrm{AiC}} = 1/s_{\mathrm{ver}}$. Finally,

$$\mathbb{E}[\tau_{\mathrm{AiC}}] \;=\; p_{\mathrm{AiC}} \sum_{n \in \mathbb{N}} n \cdot (1 - p_{\mathrm{AiC}})^{n-1} \;=\; \frac{1}{p_{\mathrm{AiC}}} \;=\; \frac{1}{s_{\mathrm{ver}}} \,.$$

**Sub-optimality.**  The mass that the AiC sampling rule assigns to the ground truth set $\mathcal{S}^\star$ is given by

$$
\begin{aligned}
\nu_{\mathrm{AiC}}(\mathcal{S}^\star) \;&=\; \nu_{AiC}(\mathcal{S}^\star \cap \widehat{\mathcal{S}}) + \nu_{\mathrm{AiC}}(\mathcal{S}^\star \setminus \widehat{\mathcal{S}}) \\
&=\; \frac{1}{s_{r^\star}} \mu(\mathcal{S}^\star \cap \widehat{\mathcal{S}}) \\
&=\; \frac{s}{s_{\mathrm{ver}}} \cdot \mathrm{TPR} \,,
\end{aligned}
\tag{18}
$$

where (18) follows from Appendix E. Hence, we have

$$
\begin{aligned}
\nu_{\mathrm{AiC}}(\mathcal{S}^\star) - \mu(\mathcal{S}^\star) \;&=\; \frac{s}{s_{\mathrm{ver}}} \cdot \mathrm{TPR} - s_{r^\star} \\
&=\; \frac{s}{s_{\mathrm{ver}}} \cdot \mathrm{TPR} - 1 + 1 - s_{r^\star} \\
&\overset{(16)}{=}\; \frac{s_{r^\star} \cdot \mathrm{TPR} - (s_{r^\star} \cdot \mathrm{TPR} + (1 - s_{r^\star}) \cdot \mathrm{FPR})}{s_{\mathrm{ver}}} + (1 - s_{r^\star}) \\
&=\; \frac{1 - s_{r^\star}}{s_{\mathrm{ver}}} (s_{\mathrm{ver}} - \mathrm{FPR}) \\
&\overset{(16)}{=}\; \frac{1 - s_{r^\star}}{s_{\mathrm{ver}}} \Big( (s_{r^\star} \cdot \mathrm{TPR} + (1 - s_{r^\star}) \cdot \mathrm{FPR}) - \mathrm{FPR} \Big)
\end{aligned}
$$

$$= \frac{1}{s_{\text{ver}}} \cdot s_{r^\star}(1 - s_{r^\star}) \cdot J \,. \tag{19}$$

Next, note that

$$\begin{aligned}
\text{SubOpt(AiC)} &\overset{(3)}{=} \nu^\star(\mathcal{S}^\star) - \nu_{\text{AiC}}(\mathcal{S}^\star) \\
&= \nu^\star(\mathcal{S}^\star) - \mu(\mathcal{S}^\star) + \mu(\mathcal{S}^\star) - \nu_{\text{AiC}}(\mathcal{S}^\star) \\
&= \text{OTC}(\beta) - \frac{1}{s_{\text{ver}}} \cdot s_{r^\star}(1 - s_{r^\star}) \cdot J \,,
\end{aligned} \tag{20}$$

where (20) follows by noting that $\nu^\star(\mathcal{S}^\star) = (1 \wedge m_\beta(s_{r^\star}))$ (using (2)), followed by using Theorem 3.1, and finally combining it with (19).

- Large coverage $- \beta > \frac{1}{s_{r^\star}}$: In this regime, we have $\text{OTC}(\beta) = 1 - s_{r^\star}$, and hence, we have:

$$\text{SubOpt(AiC)} = \text{OTC}(\beta) \left( 1 - \frac{s_{r^\star}}{s_{\text{ver}}} \cdot J \right) \,.$$

- Small coverage $- \beta \leq \frac{1}{s_{r^\star}}$: In this regime, the optimal transport cost is increasing in $\beta$, and is given by $\text{OTC}(\beta) = \sqrt{s_{r^\star}(1 - s_{r^\star})(\beta - 1)}$, and hence, we obtain:

$$\text{SubOpt(AiC)} = \text{OTC}(\beta) \cdot \left( 1 - \frac{1}{s_{\text{ver}}} \sqrt{\frac{s_{r^\star}(1 - s_{r^\star})}{\beta - 1}} \cdot J \right) \,.$$

## G   AIC CONSTRAINT VIOLATION (PROOF OF THEOREM 3.3)

Note that

$$\nu_{\text{AiC}}(d\mathbf{z}) = \mu(d\mathbf{z} \mid \widehat{\mathcal{S}}) \overset{(16)}{=} \frac{1}{s_{\text{ver}}} \cdot \mu(d\mathbf{z} \cap \widehat{\mathcal{S}}) = \frac{\mathbb{1}\{\mathbf{z} \in \widehat{\mathcal{S}}\}}{s_{\text{ver}}} \cdot \mu(d\mathbf{z}) \,.$$

Hence, we have

$$\chi^2(\mu \| \nu_{\text{AiC}}) = \int_{\mathcal{Y}} \left( \frac{d\nu_{\text{AiC}}}{d\mu} \right)^2 d\mu - 1 = \int_{\widehat{\mathcal{S}}} \left( \frac{1}{s_{\text{ver}}} \right)^2 - 1 = \frac{1}{s_{\text{ver}}} - 1 \,,$$

which implies that based on our coverage constraint, we must have $\beta \geq \frac{1}{s_{\text{ver}}}$ .

## H   SMC RESIDUAL MEASURE (PROOF OF THEOREM 3.4)

Let us define the *minimum* measure $\lambda \triangleq (\mu \wedge \nu)$. Furthermore, let us assume that $m_\beta(s_r) \geq s_r$; the complementary case follows analogously. Based on the closed-form expression for the Radon-Nikodym derivative of the target-to-proposal measures stated in (2), we have

$$\lambda = \left( 1 \wedge \frac{d\nu}{d\mu} \right) \cdot \mu = \mu \mid_{\mathcal{S}} + \frac{1 - m_\beta(s_r)}{1 - s_r} \cdot \mu \mid_{\overline{\mathcal{S}}} \,, \tag{21}$$

which gives

$$\nu - \lambda = (p - 1) \cdot \mu \mid_{\mathcal{S}} + (q - q)\mu \mid_{\overline{\mathcal{S}}} \,, \tag{22}$$

where we have set

$$p \triangleq \left( \frac{1}{s_r} \wedge \frac{m_\beta(s_r)}{s_r} \right) \,, \qquad \text{and} \qquad q \triangleq \left( \frac{1 - m_\beta(s_r)}{1 - s_r} \vee 0 \right) \,.$$

Furthermore,

$$\lambda(\mathcal{Y}) \overset{(21)}{=} s_r + \frac{1 - m_\beta(s_r)}{1 - s_r} \cdot (1 - s_r)$$

$$= 1 - (m_\beta(s_r) - s_r)$$

$$= 1 - s_r \left( \left( \frac{m_\beta(s_r)}{s_r} \wedge \frac{1}{s_r} \right) - 1 \right)$$

$$= 1 - s_r(p-1) . \tag{23}$$

Finally, we have

$$\mu_{\mathrm{res}} \overset{(22)-(23)}{=} \frac{(p-1)\mu(\cdot \cap \mathcal{S})}{s_r(p-1)} = \mu(\cdot \mid \mathcal{S}) .$$

## I   SRS / SMC COMPUTATIONAL COMPLEXITY (PROOF OF THEOREM 3.5)

**Computational complexity of SRS:** We have

$$\mathbb{P}\big(\mathbf{Z} = \mathbf{Y} \mid \mathbf{Y} \sim \pi_{\mathrm{ref}}(\cdot \mid \mathbf{x})\big)$$

$$= \mathbb{P}\big(\mathbf{Z} = \mathbf{Y} \mid \mathbf{Y} \sim \pi_{\mathrm{ref}}(\cdot \mid \mathbf{x}) , \ \mathbf{Y} \in \widehat{\mathcal{S}}\big) \cdot \underbrace{\mathbb{P}\big(\mathbf{Y} \in \widehat{\mathcal{S}}\big)}_{\overset{(16)}{=} s_{\mathrm{ver}}}$$

$$\quad + \mathbb{P}\big(\mathbf{Z} = \mathbf{Y} \mid \mathbf{Y} \sim \pi_{\mathrm{ref}}(\cdot \mid \mathbf{x}) , \ \mathbf{Y} \notin \widehat{\mathcal{S}}\big) \cdot \mathbb{P}\big(\mathbf{Y} \notin \widehat{\mathcal{S}}\big)$$

$$= s_{\mathrm{ver}} + \mathbb{P}\left( \mathbf{Z} = \mathbf{Y} \mid \mathbf{Y} \sim \pi_{\mathrm{ref}}(\cdot \mid \mathbf{x}) , \ \mathbf{Y} \notin \widehat{\mathcal{S}}, \ \frac{1}{M}\widehat{\eta}(\mathbf{Y}) \geq U , \ U \sim \mathrm{Unif}[0,1] \right)$$

$$\qquad \times \mathbb{P}\left( \frac{1}{M}\widehat{\eta}(\mathbf{Y}) \geq U , \ U \sim \mathrm{Unif}[0,1] \mid \mathbf{y} \sim \pi_{\mathrm{ref}}(\cdot \mid \mathbf{x}) , \ \mathbf{Y} \notin \widehat{\mathcal{S}} \right)$$

$$\qquad + \underbrace{\mathbb{P}\left( \mathbf{Z} = \mathbf{Y} \mid \mathbf{Y} \sim \pi_{\mathrm{ref}}(\cdot \mid \mathbf{x}) , \ \mathbf{Y} \notin \widehat{\mathcal{S}}, \ \frac{1}{M}\widehat{\eta}(\mathbf{Y}) < U , \ U \sim \mathrm{Unif}[0,1] \right)}_{= 0}$$

$$\qquad \times \mathbb{P}\left( \frac{1}{M}\widehat{\eta}(\mathbf{Y}) < U , \ U \sim \mathrm{Unif}[0,1] \mid \mathbf{y} \sim \pi_{\mathrm{ref}}(\cdot \mid \mathbf{x}) , \ \mathbf{Y} \notin \widehat{\mathcal{S}} \right)$$

$$= s_{\mathrm{ver}} + (1 - s_{\mathrm{ver}}) \cdot \frac{s_{\mathrm{ver}}}{\big(1 \wedge m_\beta(s_{\mathrm{ver}})\big)} \cdot \left( 0 \vee \frac{1 - m_\beta(s_{\mathrm{ver}})}{1 - s_{\mathrm{ver}}} \right)$$

$$= s_{\mathrm{ver}} + \frac{s_{\mathrm{ver}}}{\big(1 \wedge m_\beta(s_{\mathrm{ver}})\big)} - s_{\mathrm{ver}}$$

$$= \frac{s_{\mathrm{ver}}}{\big(1 \wedge m_\beta(s_{\mathrm{ver}})\big)} .$$

Finally, denoting $p_{\mathrm{SRS}} \triangleq \mathbb{P}\big(\mathbf{Z} = \mathbf{Y} \mid \mathbf{Y} \sim \pi_{\mathrm{ref}}(\cdot \mid \mathbf{x})\big)$, we have

$$\mathbb{E}[\tau_{\mathrm{SRS}}] = p_{\mathrm{SRS}} \sum_{n \in \mathbb{N}} n \cdot (1 - p_{\mathrm{SRS}})^{n-1} = \frac{1}{p_{\mathrm{SRS}}} = \frac{\big(1 \wedge m_\beta(s_{\mathrm{ver}})\big)}{s_{\mathrm{ver}}} .$$

**Computational complexity of SMC:** First, note that SMC's probability of acceptance for the first proposal is given by

$$\mathbb{P}\big(\mathbf{Z} = \mathbf{Y}\big) = \mathbb{P}\big(\widehat{\eta}(\mathbf{Y}) \geq U \mid U \sim \mathrm{Unif}[0,1]\big)$$

$$= \underbrace{\mathbb{P}\big(\widehat{\eta}(\mathbf{Y}) \geq U \mid U \sim \mathrm{Unif}[0,1] , \ \mathbf{Y} \in \widehat{\mathcal{S}}\big)}_{= 1} \cdot \mathbb{P}\big(\mathbf{Y} \in \widehat{\mathcal{S}}\big)$$

$$\qquad + \mathbb{P}\big(\widehat{\eta}(\mathbf{Y}) \geq U \mid U \sim \mathrm{Unif}[0,1] , \ \mathbf{Y} \notin \widehat{\mathcal{S}}\big) \cdot \mathbb{P}\big(\mathbf{Y} \notin \widehat{\mathcal{S}}\big)$$

$$\overset{(16)}{=} s_{\mathrm{ver}} + \left( 0 \vee \frac{1 - m_\beta(s_{\mathrm{ver}})}{1 - s_{\mathrm{ver}}} \right) \cdot \big(1 - s_{\mathrm{ver}}\big)$$

$$= 1 - \big(0 \vee m_\beta(s_{\mathrm{ver}}) - s_{\mathrm{ver}}\big) . \tag{24}$$

We have

$$\mathbb{E}[\tau_{\mathrm{SMC}}] = \mathbb{P}\Big(\text{first proposal accepted}\Big) \cdot 1$$

$$+ \left(1 + \sum_{n \in \mathbb{N}} n\mathbb{P}\Big(n^{\mathrm{th}} \text{ proposal is accepted}\Big)\right) \cdot \mathbb{P}\Big(\text{first proposal is rejected}\Big)$$

$$\stackrel{(24)}{=} \Big(1 - \big(0 \vee m_\beta(s_{\mathrm{ver}}) - s_{\mathrm{ver}}\big)\Big) + \left(1 + \sum_{n \in \mathbb{N}} n s_{\mathrm{ver}}(1 - s_{\mathrm{ver}})^{n-1}\right) \cdot \big((1 \wedge m_\beta(s_{\mathrm{ver}})) - s_{\mathrm{ver}}\big)$$

$$= \frac{1}{s_{\mathrm{ver}}} \cdot \big(1 \wedge m_\beta(s_{\mathrm{ver}})\big) .$$

## J   SRS / SMC SUB-OPTIMALITY (PROOF OF THEOREM 3.6)

SRS is a transport plan in $\mathcal{M}(\mu, \widehat{\nu})$ and SMC is designed from the *optimal* transport plan from $\mu$ to $\widehat{\nu}$, where we denote the optimal distribution induced by the *estimated reward*, i.e., $\widehat{\nu} \triangleq \mathrm{Law}(\mathbf{Z} \mid \mathbf{x})$ where $\mathbf{Z} \sim \widehat{\pi}(\cdot \mid \mathbf{x})$, and we define $\widehat{\pi}(\cdot \mid \mathbf{x}) \triangleq \arg\max_{\pi(\cdot \mid \mathbf{x}) \in \Pi(\beta \mid \mathbf{x})} \mathbb{E}_{\mathbf{y} \sim \pi(\cdot \mid \mathbf{x})}[\widehat{r}(\mathbf{y}, \mathbf{x})]$. Consequently, SRS and SMC sample from the same distribution $\widehat{\nu}$; our sub-optimality analysis will quantify the discrepancy induced as a result of sampling from $\widehat{\nu}$ instead of $\nu^\star$. The key in our analysis is to decompose the sub-optimality into two terms: an optimal transport cost (OTC) term, and a policy improvement (PI) term. This leads to sub-optimality having three distinct regimes, which we discuss next. Note that for $\mathfrak{A} \in \mathrm{SRS}, \mathrm{SMC}$,

$$\mathrm{SubOpt}(\mathfrak{A}) = \nu^\star(\mathcal{S}^\star) - \nu_{\mathfrak{A}}(\mathcal{S}^\star) = \nu^\star(\mathcal{S}^\star) - \widehat{\nu}(\mathcal{S}^\star) = \underbrace{\nu^\star(\mathcal{S}^\star) - \mu(\mathcal{S}^\star)}_{=\ \mathrm{OTC}} - \underbrace{\widehat{\nu}(\mathcal{S}^\star) - \mu(\mathcal{S}^\star)}_{=\ \mathrm{PI}} .$$

The mass that $\widehat{\nu}$ assigns on $\mathcal{S}^\star$ can be expanded using the Radon-Nikodym derivative in (2) as follows.

$$\widehat{\nu}(\mathcal{S}^\star) = \left(\frac{1}{s_{\mathrm{ver}}} \wedge \frac{m_\beta(s_{\mathrm{ver}})}{s_{\mathrm{ver}}}\right) \cdot \mu\big(\mathcal{S}^\star \cap \widehat{\mathcal{S}}\big) + \left(\frac{1 - m_\beta(s_{\mathrm{ver}})}{1 - s_{\mathrm{ver}}} \vee 0\right) \cdot \mu(\mathcal{S}^\star \setminus \widehat{\mathcal{S}})$$

$$= \left(\frac{1}{s_{\mathrm{ver}}} \wedge \frac{m_\beta(s_{\mathrm{ver}})}{s_{\mathrm{ver}}}\right) \cdot \mathrm{TP} + \left(\frac{1 - m_\beta(s_{\mathrm{ver}})}{1 - s_{\mathrm{ver}}} \vee 0\right) \cdot \mathrm{FN}$$

$$\stackrel{(17)}{=} \underbrace{\left(\frac{1}{s_{\mathrm{ver}}} \wedge \frac{m_\beta(s_{\mathrm{ver}})}{s_{\mathrm{ver}}}\right)}_{\triangleq\ p_{\mathrm{ver}}} \cdot s_{r^\star} \cdot \mathrm{TPR} + \underbrace{\left(\frac{1 - m_\beta(s_{\mathrm{ver}})}{1 - s_{\mathrm{ver}}} \vee 0\right)}_{\triangleq\ q_{\mathrm{ver}}} \cdot s_{r^\star} \cdot \big(1 - \mathrm{TPR}\big) . \quad (25)$$

Furthermore, expanding PI, we have

$$\widehat{\nu}(\mathcal{S}^\star) - \mu(\mathcal{S}^\star) \stackrel{(25)}{=} s_{r^\star}\Big((p_{\mathrm{ver}} - 1)\mathrm{TPR} + (q_{\mathrm{ver}} - 1)(1 - \mathrm{TPR})\Big)$$

$$= s_{r^\star}\left(\left(\frac{1 - s_{\mathrm{ver}}}{s_{\mathrm{ver}}} \wedge \frac{m_\beta(s_{\mathrm{ver}}) - s_{\mathrm{ver}}}{s_{\mathrm{ver}}}\right) \cdot \mathrm{TPR}\right.$$

$$\left. - \left(\frac{m_\beta(s_{\mathrm{ver}}) - s_{\mathrm{ver}}}{1 - s_{r^\star}} \vee -s_{\mathrm{ver}}\right) \cdot \big(1 - \mathrm{TPR}\big)\right)$$

$$= s_{r^\star} \cdot \big(m_\beta(s_{\mathrm{ver}}) - s_{\mathrm{ver}}\big) \cdot \left(\frac{\mathrm{TPR}}{s_{\mathrm{ver}}} - \frac{1 - \mathrm{TPR}}{1 - s_{\mathrm{ver}}}\right)$$

$$= s_{r^\star} \cdot \big(m_\beta(s_{\mathrm{ver}}) - s_{\mathrm{ver}}\big) \cdot \frac{\mathrm{TPR} - s_{\mathrm{ver}}}{s_{\mathrm{ver}}(1 - s_{\mathrm{ver}})}$$

$$\stackrel{(16)}{=} s_{r^\star} \cdot \big(m_\beta(s_{\mathrm{ver}}) - s_{\mathrm{ver}}\big) \cdot \frac{\mathrm{TPR} - (s_{r^\star} \cdot \mathrm{TPR} + (1 - s_{r^\star}\mathrm{FPR}))}{s_{\mathrm{ver}}(1 - s_{\mathrm{ver}})}$$

$$= \big(m_\beta(s_{\mathrm{ver}}) - s_{\mathrm{ver}}\big) \cdot \frac{s_{r^\star}(1 - s_{r^\star})}{s_{\mathrm{ver}}(1 - s_{\mathrm{ver}})} \cdot J . \quad (26)$$

Next, based on coverage, we have the following fours cases.

**Transport regime –** $\beta \leq \left(\frac{1}{s_{r^\star}} \wedge \frac{1}{s_{\mathrm{ver}}}\right)$: In this regime, we have $\mathrm{OTC}(\beta) = \sqrt{s_{r^\star}(1 - s_{r^\star})(\beta - 1)}$, which combined with (26) gives us

$$\mathrm{SubOpt}(\mathfrak{A}) \;=\; \mathrm{OTC}(\beta)\left(1 - \sqrt{\frac{s_{r^\star}(1 - s_{r^\star})}{s_{\mathrm{ver}}(1 - s_{\mathrm{ver}})} \cdot J}\right).$$

**Policy improvement regime –** $\beta \in \left(\frac{1}{s_{\mathrm{ver}}}, \frac{1}{s_{r^\star}}\right]$: In this regime, we have $\mathrm{OTC}(\beta) = \sqrt{s_{r^\star}(1 - s_{r^\star})(\beta - 1)}$ and $m_\beta(s_{\mathrm{ver}}) - s_{\mathrm{ver}} = \sqrt{s_{\mathrm{ver}}(1 - s_{\mathrm{ver}})(\beta - 1)}$, and hence, we have

$$\mathrm{SubOpt}(\mathfrak{A}) \;=\; \sqrt{s_{r^\star}(1 - s_{r^\star})(\beta - 1)} - \sqrt{\frac{\beta - 1}{s_{\mathrm{ver}}(1 - s_{\mathrm{ver}})}} \cdot s_{r^\star}(1 - s_{r^\star}) \cdot J$$

$$=\; \mathrm{OTC}(\beta)\left(1 - \frac{1}{s_{\mathrm{ver}}}\sqrt{\frac{s_{r^\star}(1 - s_{r^\star})}{\beta - 1}} \cdot J\right).$$

**Policy improvement regime –** $\beta \in \left(\frac{1}{s_{r^\star}}, \frac{1}{s_{s_{\mathrm{ver}}}}\right]$: In this regime, $\mathrm{OTC}(\beta) = 1 - s_{r^\star}$, and hence, we have

$$\mathrm{SubOpt}(\mathfrak{A}) \;=\; (1 - s_{r^\star}) - \sqrt{\frac{\beta - 1}{s_{\mathrm{ver}}(1 - s_{\mathrm{ver}})}} \cdot s_{r^\star}(1 - s_{r^\star}) \cdot J$$

$$=\; \mathrm{OTC}(\beta)\left(1 - \sqrt{\frac{\beta - 1}{s_{\mathrm{ver}}(1 - s_{\mathrm{ver}})}} \cdot s_{r^\star} \cdot J\right).$$

**Saturation regime –** $\beta > \left(\frac{1}{s_{r^\star}} \wedge \frac{1}{s_{\mathrm{ver}}}\right)$: In this regime, we have $\mathrm{OTC}(\beta) = 1 - s_{r^\star}$ and $m_\beta(s_{\mathrm{ver}}) = 1$, and it can be readily verified that

$$\mathrm{SubOpt}(\mathfrak{A}) \;=\; \mathrm{OTC}(\beta)\left(1 - \frac{s_{r^\star}}{s_{\mathrm{ver}}} \cdot J\right).$$

## K  BoN Batch Size (Proof of Theorem 3.7)

Let us denote $a \triangleq (1 - s_{r^\star})^N$. Evaluating the $\chi^2$-divergence between the measure induced by the BoN sampling policy $\nu_{\mathrm{BoN}}$ with batch size $N + 1$, we have

$$\int_{\mathcal{Y}}\left(\frac{\mathrm{d}\nu_{\mathrm{BoN}}}{\mathrm{d}\mu}\right)^2 \mathrm{d}\mu - 1 \;=\; \int_{\mathcal{S}}\left(\frac{\mathrm{d}\nu_{\mathrm{BoN}}}{\mathrm{d}\mu}\right)^2 \mathrm{d}\mu + \int_{\overline{\mathcal{S}}}\left(\frac{\mathrm{d}\nu_{\mathrm{BoN}}}{\mathrm{d}\mu}\right)^2 \mathrm{d}\mu - 1$$

$$=\; \frac{1}{s_{r^\star}}\left(1 - (1 - s_{r^\star})a\right)^2 + (1 - s_{r^\star})a^2 - 1 \qquad (27)$$

$$=\; \frac{1}{s_{r^\star}}\left(1 + (1 - s_{r^\star})^2 a^2 - 2(1 - s_{r^\star})a\right) + (1 - s_{r^\star})a^2 - 1$$

$$=\; a^2 \cdot \frac{1 - s_{r^\star}}{s_{r^\star}} - 2a\frac{1 - s_{r^\star}}{s_{r^\star}} + \frac{1 - s_{r^\star}}{s_{r^\star}}$$

$$=\; \left(\frac{1 - s_{r^\star}}{s_{r^\star}}\right)(a - 1)^2,$$

where (27) follows from Lemma D.1. Hence,

$$\chi^2\left(\nu_{\mathrm{BoN}}\|\mu\right) \;=\; \left(\frac{1 - s_{r^\star}}{s_{r^\star}}\right) \cdot \left(1 - (1 - s_{r^\star})^N\right)^2. \qquad (28)$$

Note that $\chi^2\left(\nu_{\mathrm{BoN}}\|\mu\right) \leq (1 - s_{r^\star})/s_{r^\star}$, and hence we have $N_{\max} = +\infty$ for any $\beta > (1 - s_{r^\star})/s_{r^\star}$. The regime $\beta \in (s_{r^\star}(1 - s_{r^\star}), \frac{1 - s_{r^\star}}{s_{r^\star}}]$ follows from bounding (28) by $\beta - 1$. The proof completes by noting that $\chi^2\left(\nu_{\mathrm{BoN}}\|\mu\right)$ is lower bounded by $s_{r^\star}(1 - s_{r^\star})$, which is obtained by setting $N = 1$ in (28).

## L  BoN Sub-optimality (Proof of Theorem 3.8)

From (3), we have

$$
\begin{aligned}
\mathrm{SubOpt}(\mathrm{BoN}) &= \nu^\star(\mathcal{S}^\star) - \nu_{\mathrm{BoN}}(\mathcal{S}^\star) \\
&= \nu^\star(\mathcal{S}^\star) - \left(1 - (1 - s_{r^\star})^{N+1}\right) && (29) \\
&= (1 \wedge m_\beta(s_{r^\star})) - s_{r^\star} + (1 - s_{r^\star}) + (1 - s_{r^\star})^{N+1} && (30) \\
&= (1 - s_{r^\star})^{N+1} - (0 \vee 1 - m_\beta(s_r^\star)),
\end{aligned}
$$

where (29) follows from Lemma D.1 and (30) follows from Theorem 2.1.

## M  BRS Batch Size (Proof of Theorem 3.9)

Let us set $a = (1 - \frac{1}{M})^{-1}$. We have

$$
\begin{aligned}
\chi^2(\nu_{\mathrm{BRS}}\|\mu) &= \int \left(\frac{\mathrm{d}\nu_{\mathrm{BRS}}}{\mathrm{d}\mu}\right)^2 \mathrm{d}\mu - 1 \\
&= \int \left((1 - a^{-N})\frac{\mathrm{d}\nu^\star}{\mathrm{d}\mu}(\mathbf{y}) + a^{-N}\right)^2 \mu(\mathrm{d}\mathbf{y}) - 1 && (31) \\
&= \int (1 - a^{-N})^2 \left(\frac{\mathrm{d}\nu^\star}{\mathrm{d}\mu}\right)^2 \mu(\mathrm{d}\mathbf{y}) + \int a^{-2N}\mu(\mathrm{d}\mathbf{y}) \\
&\qquad + 2\int a^{-N}(1 - a^{-N})\frac{\mathrm{d}\nu^\star}{\mathrm{d}\mu}\mu(\mathrm{d}\mathbf{y}) - 1 \\
&= \int_{\mathcal{S}^\star}(1 - a^{-N})^2 \left(\frac{\mathrm{d}\nu^\star}{\mathrm{d}\mu}\right)^2 \mu(\mathrm{d}\mathbf{y}) + \int_{\overline{\mathcal{S}}^\star}(1 - a^{-N})^2 \left(\frac{\mathrm{d}\nu^\star}{\mathrm{d}\mu}\right)^2 \mu(\mathrm{d}\mathbf{y}) \\
&\qquad + a^{-2N} + 2a^{-N}(1 - a^{-N}) - 1 \\
&= \int_{\mathcal{S}}(1 - a^{-N})^2 \left(\frac{1}{s_{r^\star}} \wedge \frac{m_\beta(s_{r^\star})}{s_{r^\star}}\right)\nu^\star(\mathrm{d}\mathbf{y}) + a^{-N}\left(a^{-N} + 2 - 2a^{-N}\right) - 1 && (32) \\
&\qquad + \int_{\overline{\mathcal{S}}^\star}(1 - a^{-N})^2 \cdot \left(0 \vee \frac{1 - m_\beta(s_{r^\star})}{1 - s_{r^\star}}\right)\nu(\mathrm{d}\mathbf{y}) \\
&= (1 - a^{-N})^2 \cdot \frac{1}{s_{r^\star}}\left(1 \wedge m_\beta(s_{r^\star})\right) + (1 - s^{-N})^2 \cdot \frac{1}{1 - sr^\star} \cdot \left(0 \vee 1 - m_\beta(s_{r^\star})\right)^2 \\
&\qquad + a^{-N}(2 - a^{-N}) - 1 \\
&= (1 - a^{-N})^2 \cdot \underbrace{\left(\frac{1}{s_{r^\star}}(1 \wedge m_\beta(s_{r^\star}))^2 + \frac{1}{1 - s_{r^\star}}(0 \vee 1 - m_\beta(s_{r^\star}))^2\right)}_{\triangleq C} \\
&\qquad + \underbrace{a^{-N}}_{\triangleq t}(2 - a^{-N}) - 1 \\
&= (1 - t)^2 \cdot C - (1 - 2t + t^2) \\
&= (1 - t)\,(C - 1),
\end{aligned}
$$

where (31) follows from Lemma D.2 and (32) follows from Theorem 2.1. Next, investigating $C$, we have the following two cases.

Case A: $(1 \wedge m_\beta(s_{r^\star})) = m_\beta(s_{r^\star})$: In this case, denoting $d \triangleq \sqrt{s_{r^\star}(1 - s_{r^\star})(\beta - 1)}$, we have

$$
\begin{aligned}
C &= \frac{(s_{r^\star} + d)^2}{s_{r^\star}} + \frac{(1 - s_{r^\star} - d)^2}{1 - s_{r^\star}} \\
&= 1 + d^2 \left(\frac{1}{s_{r^\star}} + \frac{1}{1 - s_{r^\star}}\right)
\end{aligned}
$$

$$
\begin{aligned}
&= 1 + s_{r^\star}(1 - s_{r^\star})(\beta - 1)\left(\frac{1}{s_{r^\star}} + \frac{1}{1 - s_{r^\star}}\right) \\
&= a + (\beta - 1)(1 - s_{r^\star} + s_{r^\star}) \\
&= \beta .
\end{aligned}
$$

Case B: $(1 \wedge m_\beta(s_{r^\star})) = 1$: In this case, we have $C = \frac{1}{s_{r^\star}}$. Furthermoremore, leveraging the condition in this case that $m_\beta(s_{r^\star}) \geq 1$, we find that $\beta \geq \frac{1}{s_{r^\star}}$, consequently establishing that $C \leq \beta$. Our proof concludes by noting that $(1 - a^{-N})^2 \leq 1$ for any $N \in \mathbb{N}$.

## N  BRS SUB-OPTIMALITY (PROOF OF THEOREM 3.10)

From (3) we obtain that

$$
\begin{aligned}
\mathrm{SubOpt}(\mathrm{BRS}) &= \nu^\star(\mathcal{S}^\star) - \nu_{\mathrm{BRS}}(\mathcal{S}^\star) \\
&= \left(1 \wedge m_\beta(s_{r^\star})\right) - \nu_{\mathrm{BRS}}(\mathcal{S}^\star) &&(33) \\
&= \left(1 - \frac{1}{M}\right)^N \cdot \left(1 \wedge m_\beta(s_{r^\star})\right) - \left(1 - \frac{1}{M}\right)^N \cdot s_{r^\star} &&(34) \\
&= \left(1 - \frac{1}{M}\right)^N \cdot \left(1 - s_{r^\star} \wedge m_\beta(s_{r^\star}) - s_{r^\star}\right) \\
&= \mathrm{OTC}(\beta) \cdot \left(1 - \frac{1}{M}\right)^N , &&(35)
\end{aligned}
$$

where (33) follows from Theorem 2.1, (34) follows from Lemma D.2, and finally, (35) follows from Lemma 3.1.

## O  BATCHED SAMPLING ALGORITHMS WITH APPROXIMATE VERIFIERS

In this section, we extend the sub-optimality analyses for the batched sampling algorithms BoN and BRS to settings where we only have access to an approximate verifier, captured through the set membership oracle $\widehat{\mathcal{S}}$. We begin by analyzing BoN sub-optimality with access to $\widehat{\mathcal{S}}$, and subsequently state the same for BRS. We conclude the section discussing the different regimes of the sub-optimality–coverage plot, and which algorithm is preferred in each of these regimes. For our analyses, we decompose the sub-optimality into two components, a *sampling* error, and a *verification* error. Specifically, for any algorithm $\mathfrak{A} \in \{\mathrm{BoN},\ \mathrm{BRS}\}$, let $\widehat{\nu}_\mathfrak{A}$ denote the distribution induced by its sampling mechanism. Accordingly, we have

$$
\mathrm{SubOpt}(\mathfrak{A}) \overset{(3)}{=} \nu^\star(\mathcal{S}^\star) - \widehat{\nu}_\mathfrak{A}(\mathcal{S}^\star) = \underbrace{\nu^\star(\mathcal{S}^\star) - \nu_\mathfrak{A}(\mathcal{S}^\star)}_{\text{sampling error}} + \underbrace{\nu_\mathfrak{A}(\mathcal{S}^\star) - \widehat{\nu}_\mathfrak{A}(\mathcal{S}^\star)}_{\text{verification error}} . \tag{36}
$$

We have the following theorem for BoN sub-optimality.

**Theorem O.1** (BoN – sub-optimality with approximate verifiers)**.** *The sub-optimality of the BoN sampling algorithm with access to an approximate membership oracle $\widehat{\mathcal{S}}$ is given by*

$$
\mathrm{SubOpt}(\mathrm{BoN}) = (1 - s_{r^\star})\left(1 - \frac{s_{r^\star}}{s_{\mathrm{ver}}}\left(1 - (1 - s_{\mathrm{ver}})^N\right)J\right) - \left(0 \vee 1 - m_\beta(s_{r^\star})\right) .
$$

*Proof.* From (36), we observe that it is sufficient to evaluate the verification error, since the sampling error has already been analyzed in Theorem 3.8. We have

$$
\begin{aligned}
&\nu_{\mathrm{BoN}}(\mathcal{S}^\star) - \widehat{\nu}_{\mathrm{BoN}}(\mathcal{S}^\star) \\
&= \nu_{\mathrm{BoN}}(\mathcal{S}^\star) - \left(\widehat{\nu}_{\mathrm{BoN}}(\mathcal{S}^\star \cap \widehat{\mathcal{S}}) + \widehat{\nu}_{\mathrm{BoN}}(\mathcal{S}^\star \setminus \widehat{\mathcal{S}})\right) \\
&= \left(1 - (1 - s_{r^\star})^{N+1}\right) - \left(\frac{1}{s_{\mathrm{ver}}}\left(1 - (1 - s_{\mathrm{ver}})^{N+1}\right) \cdot \mu(\mathcal{S}^\star \cap \widehat{\mathcal{S}})\right) \tag{37}
\end{aligned}
$$

$$+ (1 - s_{\mathrm{ver}})^N \mu(\mathcal{S}^\star \setminus \widehat{\mathcal{S}}) \bigg)$$

$$= \left(1 - (1 - s_{r^\star})^{N+1}\right) - \left(\frac{s_{r^\star}}{s_{\mathrm{ver}}} \left(1 - (1 - s_{\mathrm{ver}})^{N+1}\right) \cdot \mathrm{TPR}\right.$$

$$\left. + (1 - s_{\mathrm{ver}})^N \cdot s_{r^\star} \cdot (1 - \mathrm{TPR})\right)$$

$$= \left(1 - (1 - s_{r^\star})^{N+1}\right) - \left(s_{r^\star}(1 - s_{\mathrm{ver}})^N\left((1 - \mathrm{TPR})\right.\right.$$

$$\left.\left. - \frac{(1 - s_{\mathrm{ver}}) \cdot \mathrm{TPR}}{s_{\mathrm{ver}}}\right) + \frac{s_{r^\star} \cdot \mathrm{TPR}}{s_{\mathrm{ver}}}\right)$$

$$= \left(1 - (1 - s_{r^\star})^{N+1}\right) - \left(s_{r^\star}(1 - s_{\mathrm{ver}})^N\left(\frac{s_{\mathrm{ver}} - \mathrm{TPR}}{s_{\mathrm{ver}}}\right) + \frac{s_{r^\star} \cdot \mathrm{TPR}}{s_{\mathrm{ver}}}\right)$$

$$= \left(1 - (1 - s_{r^\star})^{N+1}\right) - \frac{s_{r^\star}}{s_{\mathrm{ver}}}\left((1 - s_{\mathrm{ver}})^N \cdot (s_{\mathrm{ver}} - \mathrm{TPR}) + \mathrm{TPR}\right)$$

$$\overset{(16)}{=} \left(1 - (1 - s_{r^\star})^{N+1}\right) - \frac{s_{r^\star}}{s_{\mathrm{ver}}}\left(\mathrm{TPR} - (1 - s_{r^\star})(1 - s_{\mathrm{ver}})^N \cdot J\right)$$

$$\overset{(16)}{=} \left(1 - (1 - s_{r^\star})^{N+1}\right) - \frac{s_{r^\star}}{s_{\mathrm{ver}}}\left(s_{\mathrm{ver}} + (1 - s_{r^\star}) \cdot \left(1 - (1 - s_{\mathrm{ver}})^N\right) \cdot J\right)$$

$$= \left(1 - (1 - s_{r^\star})^{N+1}\right) - \left(s_{r^\star} + \frac{s_{r^\star}(1 - s_{r^\star})}{s_{\mathrm{ver}}}\left(1 - (1 - s_{\mathrm{ver}})^N\right) \cdot J\right), \tag{38}$$

where (37) follows from Lemma D.1. The claim readily follows by combining (38) with Theorem 3.8 using (36). ∎

Next, we state the sub-optimality of the BRS algorithm with access to an approximate oracle $\widehat{\mathcal{S}}$.

**Theorem O.2.** *Let us set* $a_N \triangleq (1 - (1 - \frac{1}{M})^N)$*. The sub-optimality of the BRS algorithm with access to an approximate membership oracle* $\widehat{\mathcal{S}}$ *is specified through the following coverage regimes.*

1. **Transport regime:** *In the transport regime, characterized by the coverage constraint* $\beta \le (\frac{1}{s_{r^\star}} \wedge \frac{1}{s_{\mathrm{ver}}})$*, we have*

$$\mathrm{SubOpt}(\mathrm{BRS}) = \mathrm{OTC}(\beta)(1 - a_N) + a_N s_{r^\star}\left(\frac{1}{s_{r^\star}} m_\beta(s_{r^\star}) - \left(\frac{m_\beta(s_{\mathrm{ver}})}{s_{\mathrm{ver}}} \cdot \mathrm{TPR}\right.\right.$$
$$\left.\left. + \frac{1 - m_\beta(s_{\mathrm{ver}})}{1 - s_{\mathrm{ver}}}(1 - \mathrm{TPR})\right)\right).$$

2. **Policy improvement regime:** *We have two cases. If* $s_{\mathrm{ver}} > s_{r^\star}$*, in the policy improvement regime, characterized by the coverage constraint* $\beta \in (\frac{1}{s_{\mathrm{ver}}}, \frac{1}{s_{r^\star}}]$*, we have*

$$\mathrm{SubOpt}(\mathrm{BRS}) = \mathrm{OTC}(\beta)(1 - a_N) + a_N s_{r^\star}\left(\frac{m_\beta(s_{r^\star})}{s_{r^\star}} - \frac{\mathrm{TPR}}{s_{\mathrm{ver}}}\right).$$

*Alternatively, for* $\beta \in (\frac{1}{s_{r^\star}}, \frac{1}{s_{\mathrm{ver}}}]$*, we have*

$$\mathrm{SubOpt}(\mathrm{BRS}) = \mathrm{OTC}(\beta)(1 - a_N) + a_N s_{r^\star}\left(\frac{1}{s_{r^\star}} - \left(\frac{m_\beta(s_{\mathrm{ver}})}{s_{\mathrm{ver}}} \cdot \mathrm{TPR}\right.\right.$$
$$\left.\left. + \frac{1 - m_\beta(s_{\mathrm{ver}})}{1 - s_{\mathrm{ver}}}(1 - \mathrm{TPR})\right)\right).$$

3. **Saturation regime:** *In the saturation regime, characterized by the coverage constraint* $\beta > (\frac{1}{s_{r^\star}} \vee \frac{1}{s_{\mathrm{ver}}})$*, we have*

$$\mathrm{SubOpt}(\mathrm{BRS}) = \mathrm{OTC}(\beta)(1 - a_N) + a_N s_{r^\star}\left(\frac{1}{s_{r^\star}} - \frac{\mathrm{TPR}}{s_{\mathrm{ver}}}\right).$$

*Proof.* Similarly to Theorem O.1, we will analyze the the verification error for BRS. For clarity in presentation, let us define

$$p(s) \triangleq \left( \frac{1}{s} \wedge \frac{m_\beta(s)}{s} \right), \quad \text{and} \quad q(s) \triangleq \left( 0 \vee \frac{1 - m_\beta(s)}{1 - s} \right). \tag{39}$$

Note that

$$\begin{aligned}
&\widehat{\nu}_{\mathrm{BRS}}(\mathcal{S}^\star) \\
&= \widehat{\nu}_{\mathrm{BRS}}(\widehat{\mathcal{S}} \cap \mathcal{S}^\star) + \widehat{\nu}_{\mathrm{BRS}}(\mathcal{S}^\star \setminus \widehat{\mathcal{S}}) \\
&= \Big( a_N \cdot p(s_{\mathrm{ver}}) + (1 - a_N) \Big) \mu(\widehat{\mathcal{S}} \cap \mathcal{S}^\star) + \Big( a_N \cdot q(s_{\mathrm{ver}}) + (1 - s_N) \Big) \mu(\mathcal{S}^\star \setminus \widehat{\mathcal{S}}) \tag{40} \\
&= \Big( a_N p(s_{\mathrm{ver}}) + (1 - a_N) \Big) \cdot s_{r^\star} \cdot \mathrm{TPR} + \Big( a_N q(s_{\mathrm{ver}}) + (1 - a_N) \Big) \cdot s_{r^\star} \cdot (1 - \mathrm{TPR}) \\
&= a_N \cdot s_{r^\star} \Big( p(s_{\mathrm{ver}}) \cdot \mathrm{TPR} + q(s_{\mathrm{ver}}) \cdot (1 - \mathrm{TPR}) \Big) + (1 - a_N) s_{r^\star}, \tag{41}
\end{aligned}$$

where (40) follows from Lemma D.2. Furthermore, it can be readily verified that

$$\nu_{\mathrm{BRS}}(\mathcal{S}^\star) = a_N \cdot s_{r^\star} \cdot p(s_{r^\star}) + (1 - a_N) s_{r^\star}. \tag{42}$$

Combining (41) and (42), we have

$$\begin{aligned}
&\nu_{\mathrm{BRS}}(\mathcal{S}^\star) - \widehat{\nu}_{\mathrm{BRS}}(\mathcal{S}^\star) \\
&\qquad = a_N \Big( s_{r^\star} p(s_{r^\star}) - p(s_{\mathrm{ver}}) \cdot \mathrm{TPR} \cdot s_{r^\star} - q(s_{\mathrm{ver}}) \cdot (1 - \mathrm{TPR}) \cdot s_{r^\star} \Big) \\
&\qquad \stackrel{(39)}{=} a_N \Bigg( (1 \wedge m_\beta(s_{r^\star})) - \left( \frac{1}{s_{\mathrm{ver}}} \wedge \frac{m_\beta(s_{\mathrm{ver}})}{s_{\mathrm{ver}}} \right) \cdot \mathrm{TPR} \cdot s_{r^\star} \\
&\qquad\qquad\qquad - \left( 0 \vee \frac{1 - m_\beta(s_{\mathrm{ver}})}{1 - s_{\mathrm{ver}}} \right) (1 - \mathrm{TPR}) s_{r^\star} \Bigg).
\end{aligned}$$

**Transport regime:** In this regime, since both $m_\beta(a_r^\star)$ and $m_\beta(s_{\mathrm{ver}})$ are less than 1, we have

$$\begin{aligned}
&\nu_{\mathrm{BRS}}(\mathcal{S}^\star) - \widehat{\nu}_{\mathrm{BRS}}(\mathcal{S}^\star) \\
&\qquad = a_N \left( m_\beta(s_{r^\star}) - \frac{m_\beta(s_{\mathrm{ver}})}{s_{\mathrm{ver}}} \cdot s_{r^\star} \cdot \mathrm{TPR} - \frac{1 - m_\beta(s_{\mathrm{ver}})}{1 - s_{\mathrm{ver}}} \cdot (1 - \mathrm{TPR}) \cdot s_{r^\star} \right) \\
&\qquad = a_N s_{r^\star} \left( \frac{m_\beta(s_{r^\star})}{s_{r^\star}} - \left( \frac{m_\beta(s_{\mathrm{ver}})}{s_{\mathrm{ver}}} \cdot \mathrm{TPR} + \frac{1 - m_\beta(s_{\mathrm{ver}})}{1 - s_{\mathrm{ver}}} \cdot (1 - \mathrm{TPR}) \right) \right). \tag{43}
\end{aligned}$$

Finally, the result readily follows by adding the sampling error, $\mathrm{OTC}(\beta)(1 - a_N)$ proved in Theorem 3.10.

**Policy improvement regime:** This regime can be divided into two cases, one in which $\beta \in (1/s_{\mathrm{ver}}, 1/s_{r^\star}]$, and the second in which $\beta \in (1/s_{r^\star}, 1/s_{\mathrm{ver}}]$. In the first regime, the result readily follows by replacing $m_\beta(s_{\mathrm{ver}}) = 1$ in (43), and adding the sampling error $\mathrm{OTC}(\beta)(1 - a_N)$. In the second regime, the result readily follows by replacing $m_\beta(s_{r^\star}) = 1$ in (43), and adding $\mathrm{OTC}(\beta)(1 - a_N)$, the sampling error.

**Saturation regime:** In this regime, both $m_\beta(s_{r^\star}) = 1$ and $m_\beta(s_{\mathrm{ver}}) = 1$, and the result readily follows by replacing these values in (43) and adding the sampling error $\mathrm{OTC}(\beta)(1 - a_N)$. This concludes our proof. ∎

**Interpreting the results.** In Theorem O.1, we note that as $N$ goes to $+\infty$, the BoN sampling error decays to 0 (and potentially becomes negative, depending on whether the mass put on $\mathcal{S}^\star$ by the skyline policy is less than 1). However, the estimation error saturates at $\mathrm{OTC}(\beta)(1 - \frac{s_{r^\star}}{s_{\mathrm{ver}}} J)$, as we had observed for AiC. This is intuitive, since the verification error is entirely controlled by verifier inaccuracies, and does not depend on the design of the sampling algorithm. Similarly, from Theorem O.2, we observe a similar trend — the sampling error is driven down to 0 as the batch size $N \to +\infty$, while the verification error stagnates at $\mathrm{OTC}(\beta) \cdot (1 - \frac{s_{r^\star}}{s_{\mathrm{ver}}} J)$ under the saturation regime.

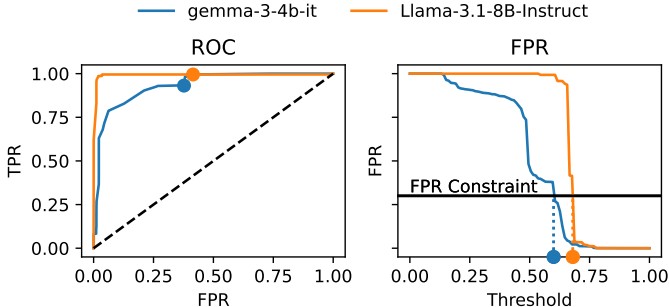

Figure 6: ROC estimated based on generations from `google/gemma-3-4b-it` (left), and `meta-llama/Llama-3.1-8B-Instruct` (right). We observe that `meta-llama/Llama-3.1-8B-Instruct` has a larger area under the curve (AUC) compared to `google/gemma-3-4b-it`

## P EXTENDED EXPERIMENTS

In this section, we specify how we construct ground-truth and approximate verifiers, the models used to evaluate our algorithms, and the hyperparameters employed in our experiments. The evaluations are conducted on GSM8K (Cobbe et al., 2021), MATH500 (Hendrycks et al.), and OlympiadBench (He et al., 2024).

For GSM8K, prompts are constructed by prefixing each question with five randomly sampled training exemplars. As in (Dorner et al., 2025; Huang et al., 2025a), we then draw 10,000 responses $\mathbf{y} \sim \pi_{\mathrm{ref}}(\cdot \mid \mathbf{x})$ at temperature 1, using models from the Qwen, Deepseek, Gemma and Llama families. Specifically, we evaluate: (i) Qwen3-1.7B, (ii) Qwen3-8B, (iii) Qwen3-14B, (iv) `google/gemma-3-4b-it`, (v) `deepseek-ai/deepseek-math-7b-instruct` and (v) `meta-llama/Llama-3.1-8B-Instruct`, spanning sizes from 1.7B to 14B parameters. Generations are obtained through the `lm-eval-harness` framework (Gao et al., 2024). Verifiers are constructed in two modes: an *explicit-construction* mode and a *reward-guided* mode. For sampling, we bootstrap from the 10,000-response pool with replacement.

**Explicit verifier construction.** To construct $\mathcal{S}^\star$, we determine the ground-truth correctness of each response by extracting the predicted answer via pattern matching with `(-?[$0-9.,]2,)|(-?[0-9]+)`, and marking it correct if it matches the dataset gold label. The proposal's mass on $\mathcal{S}^\star$, denoted $s_{r^\star}$, is estimated empirically by summing the normalized `logprobs` of correct responses. For the approximate verifier $\widehat{\mathcal{S}}$, we adopt an *explicit construction* designed to validate our theoretical analysis. Specifically, we curate subsets of correct and incorrect responses into $\widehat{\mathcal{S}}$ such that both the Youden index $J$ and the proposal's mass $s_{\mathrm{ver}}$ are controlled, thereby fixing the verifier's TPR and FPR. This provides direct and interpretable control over the verifier's operating characteristics. To ensure determinism, responses are ranked in descending order of their `logprobs`. Candidates are then selected from $\mathcal{S}^\star$ and its complement $\overline{\mathcal{S}^\star}$, starting with the highest-probability responses in each set, and iteratively added until the cumulative mass matches the preset values of $J$ and $s_{\mathrm{ver}}$.

**Reward-guided verifier construction.** As a second mode of verification to score the generated responses, we employ as reward models `Skywork/Skywork-Reward-V2-Llama-3.1-8B`, which ranks $1^{\mathrm{st}}$ on the RewardBench leaderboard (Malik et al., 2025), and `nvidia/AceMath-7B-RM`, state-of-art in math reward models (Liu et al., 2025). We derive approximate verifiers by thresholding: for a prompt $\mathbf{x} \in \mathcal{X}$ and response $\mathbf{y} \in \mathcal{Y}$, a response is included in $\widehat{\mathcal{S}}$ if its reward $r_{\mathrm{sr}}(\mathbf{x}, \mathbf{y})$ exceeds a threshold $\gamma$. By varying $\gamma$, we obtain a family of verifiers with different receiver operating characteristics (ROCs). In order to show that the three-regime structure generalizes across tasks, problems and models, we show the sub-optimality of RS, SMS, and AiC as function of $\beta$ for different value of $\gamma$ in the following settings.

## P.1 GSM8K

In Section 4, we presented results using the `Qwen3-1.7B` and `Qwen3-14B` models for question 2. Here, we supplement these with additional plots for `Qwen3-8B` under the explicit-verifier setting, along with further analyses illustrating how average reward varies with generator coverage and how sub-optimality scales with computational complexity and Youden's index. Figure 7 and Figure 8 report these results for the `Qwen` model family under the sequential sampling protocol. Figures 9 and 10 provide the corresponding plots for the batched setting, i.e., BoN and BRS. Figure 11 reports analogous plots under a reward-guided verifier constructed with `Skywork/Skywork-Reward-V2-Llama-3.1-8B`, and Figure 6 shows the ROCs for `google/gemma-3-4b-it` and `meta-llama/Llama-3.1-8B-Instruct`. Figures 12 and 13 shows how the sub-optimality regimes change across different $\beta$ and $\gamma$ for both `google/gemma-3-4b-it` and `meta-llama/Llama-3.1-8B-Instruct`, respectively. Moreover, Figures 14, 15, and 16 show the same for questions 11, 14, and 4 when `nvidia/AceMath-7B-RM` is used as judge.

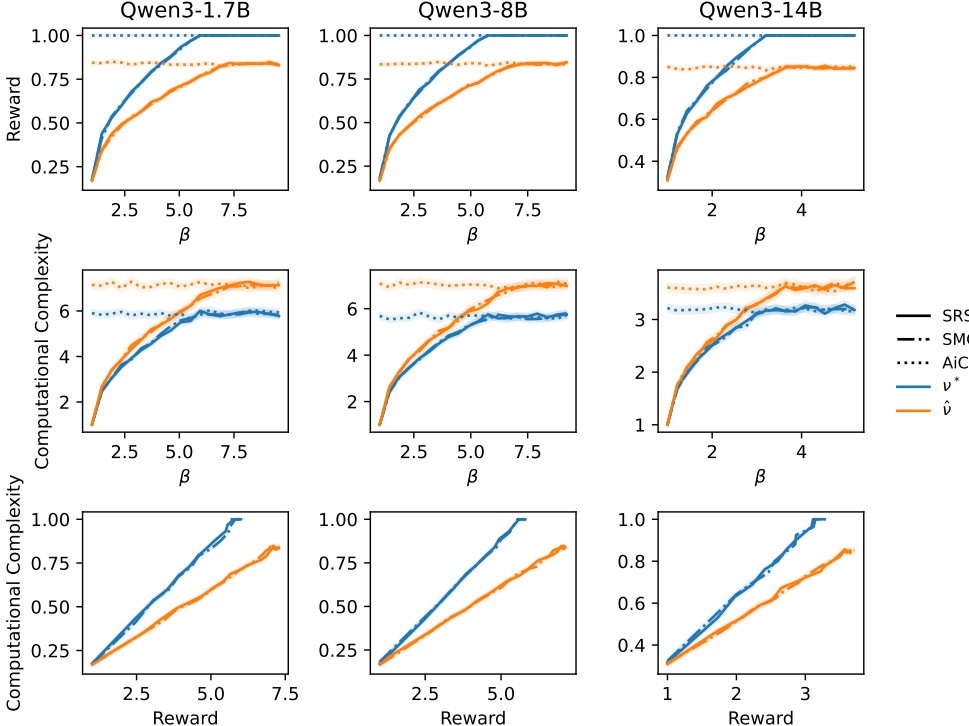

Figure 7: Plots for sample 2 of GSM8K dataset using the `Qwen` family with an *explicit verifier*: **average reward versus** $\beta$ (first row), **computational complexity versus** $\beta$ (second row), and **computational complexity versus reward** (third row). Trends predicted in Theorems 3.2 and 3.5 are observed.

## P.2 MATH500 & OLYMPIADBENCH

In order to further demonstrate that the three-regime structure generalizes across tasks, problems, and models, we report the sub-optimality of RS, SMS, and AiC as a function of $\beta$ for different values of $\gamma$ on both the MATH500 and OlympiadBench datasets. For MATH500, we use `deepseek-ai/deepseek-math-7b-instruct` as the generator model for problems 2 and 8 (Figures 20 and 21) and `meta-llama/Llama-3.1-8B-Instruct` for problem 28 (Figure 22). For OlympiadBench, we adopt `deepseek-ai/deepseek-math-7b-instruct` for sample 14. For both datasets, we use `nvidia/AceMath-7B-RM` to construct the reward-guided verifier.

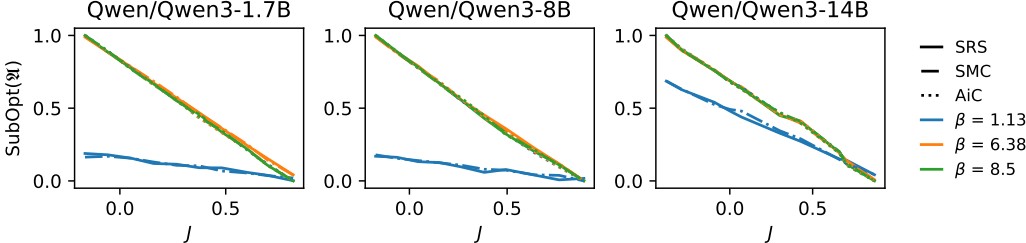

Figure 8: **Sub-optimality plotted against Youden's index** $J$ for the `Qwen` model family with an *explicit verifier*, using $\beta_T, \beta_{PI}, \beta_S$ to represent the three distinct coverage regimes. $\beta$ values are computed as $\beta_T = (0.2 \cdot \underline{\beta_{sat}} \vee 1)$, $\beta_{PI} = (\beta_T + \bar{\beta}_{sat})/2$, $\beta_S = 1.2 \cdot \bar{\beta}_{sat}$, where $\underline{\beta_{sat}} = (1/s_{r^\star} \wedge 1/s_{ver})$, and $\bar{\beta}_{sat} = (1/s_{r^\star} \vee 1/s_{ver})$.

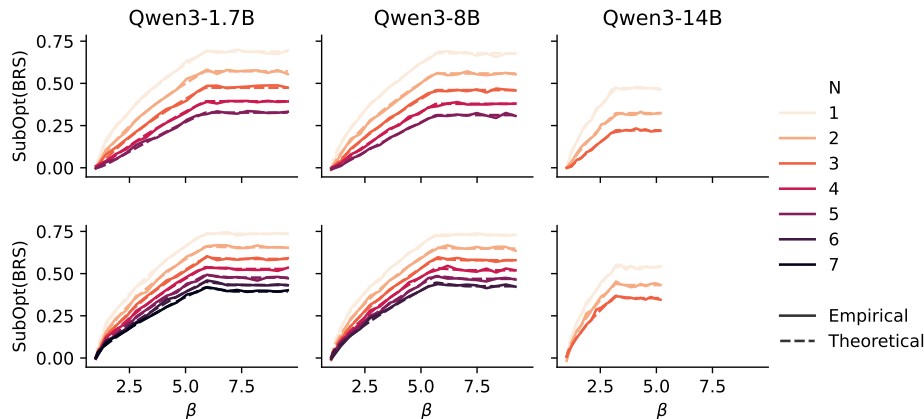

Figure 9: BRS plots for the `Qwen` family with an *explicit verifier*: **ground truth verifier** on the first row, **explicit verifier** on the second row. Plots match the theoretical findings in Theorems 3.10 and O.2. Furthermore, as $N$ increases, sub-optimality decreases.

## Q   COMPUTE AND LLM USAGE

All generations are performed in $8\times$ A6000 Nvidia GPUs with 49 gigabytes of VRAM each. LLMs have been used for (1) sharpening the write-up, (2) as a coding assistant for the experiments, and (3) verifying the correctness of some algebra in the proofs of Lemma D.1 and Theorem 3.8.

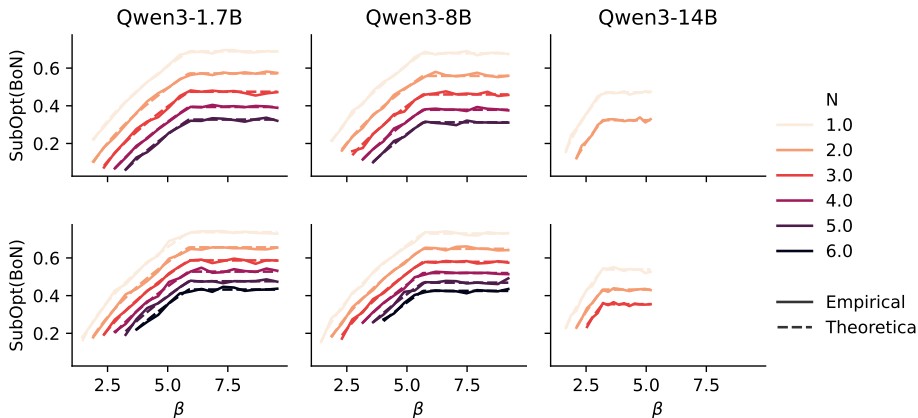

Figure 10: BoN plots for the `Qwen` family with an *explicit verifier*: **ground truth verifier** on the first row, **explicit verifier** on the second row. Plots match the theoretical findings in Theorems 3.8 and O.1. Here, we choose $N \in [\lfloor (N_{\max} \wedge \frac{1}{s}) \rfloor]$ as prescribed in Theorem 3.7 for feasibility.

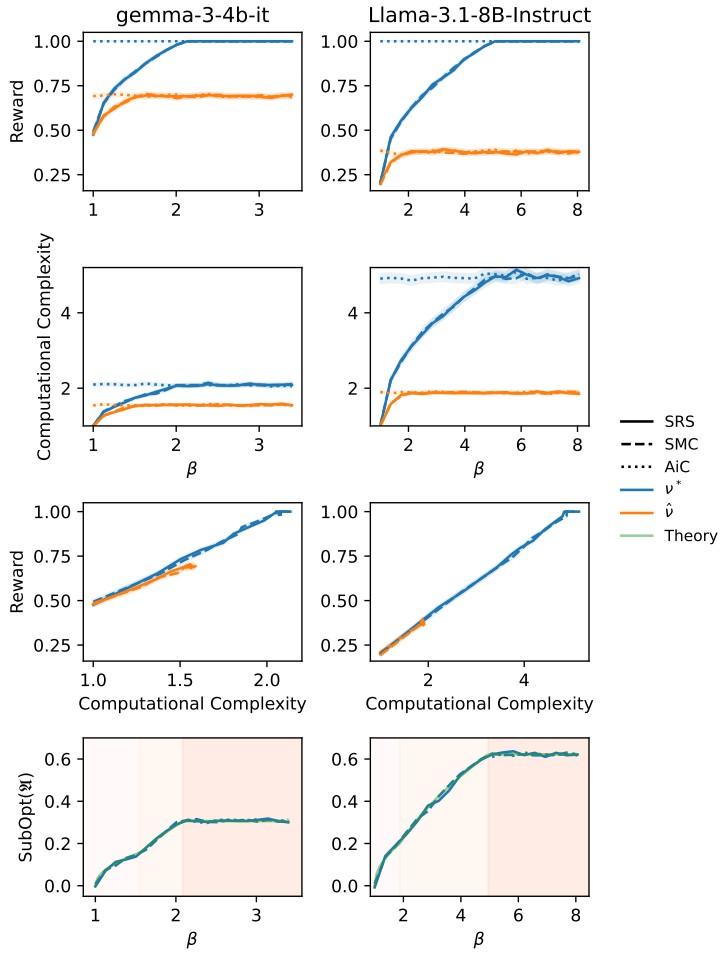

Figure 11: **Reward-guided verifier:** verifiers chosen as indicated in the ROC plot in Figure 6. We plot **reward versus** $\beta$ (first row), **computational complexity versus** $\beta$ (second row), **reward versus computational complexity** (third row), and **sub-optimality versus** $\beta$ (fourth row.)

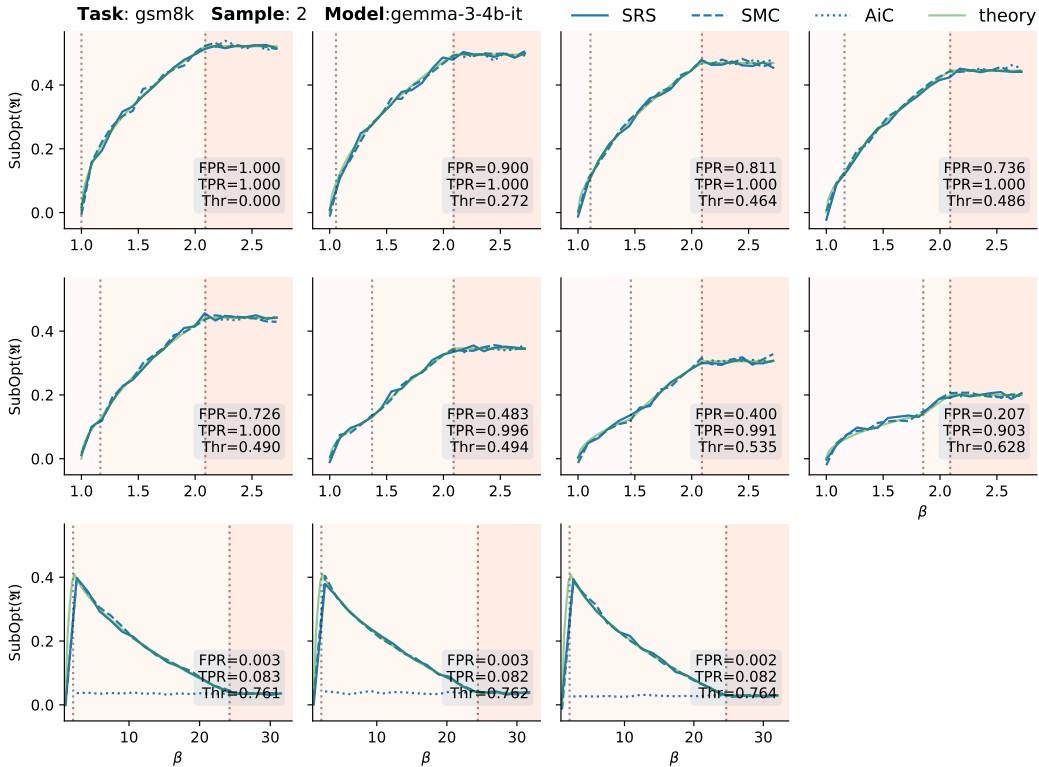

Figure 12: **Reward-guided verifiers for different** $\gamma$**:**. We plot sub-optimality versus $\beta$ as varying $\gamma$ (reported as Thr in each plot, together with FPR and TPR) for `google/gemma-3-4b-it`.

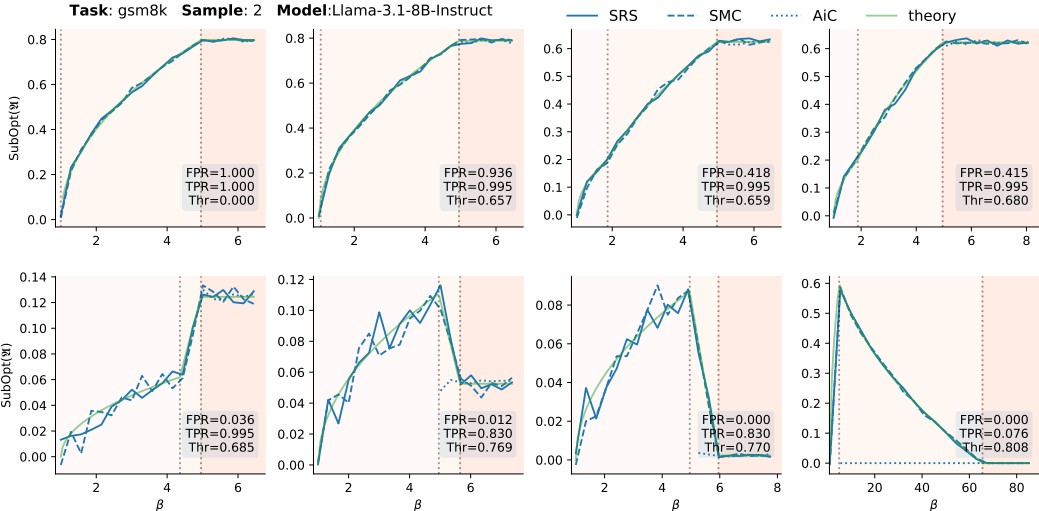

Figure 13: **Reward-guided verifiers for different** $\gamma$**:**. We plot sub-optimality versus $\beta$ as varying $\gamma$ (reported as Thr in each plot, together with FPR and TPR) for `meta-llama/Llama-3.1-8B-Instruct`.

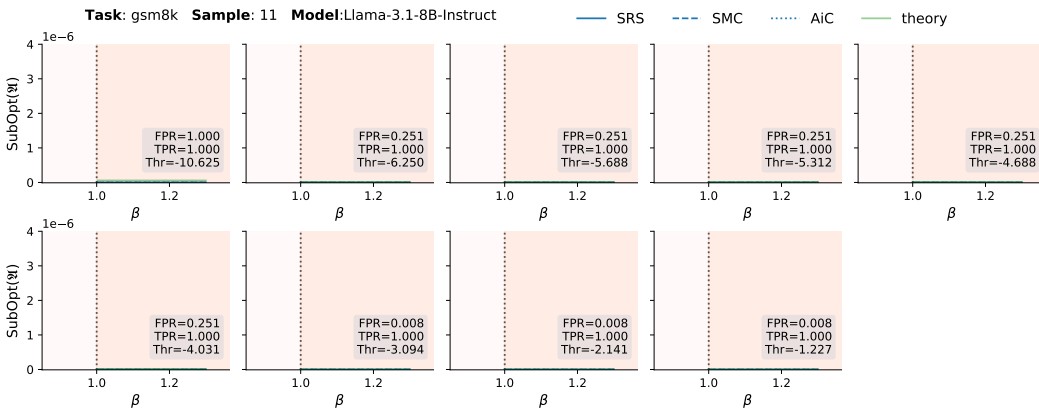

Figure 14: Sub-optimality as a function of $\beta$ for varying thresholds of the LLM-as-judge verifier, evaluated on sample 11 of the GSM8K dataset using `Llama-3.1-8B-instruct`.

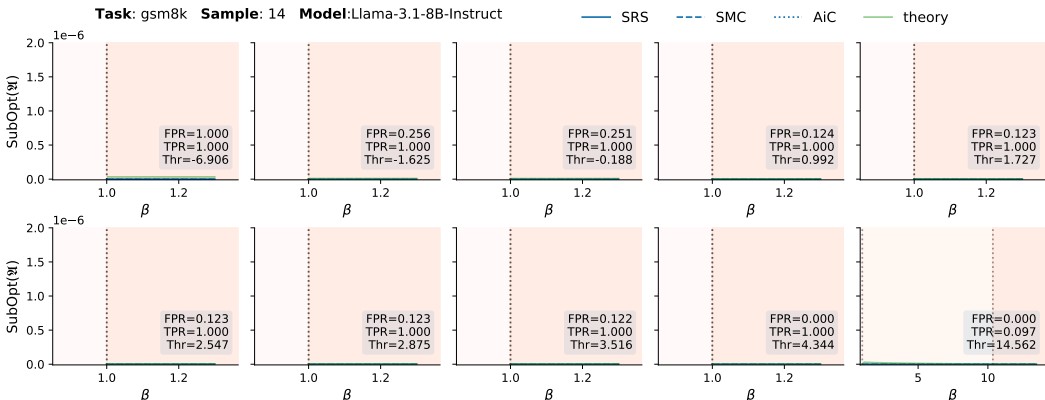

Figure 15: Sub-optimality as a function of $\beta$ for varying thresholds of the LLM-as-judge verifier, evaluated on sample 14 of the GSM8K dataset using `Llama-3.1-8B-instruct`.

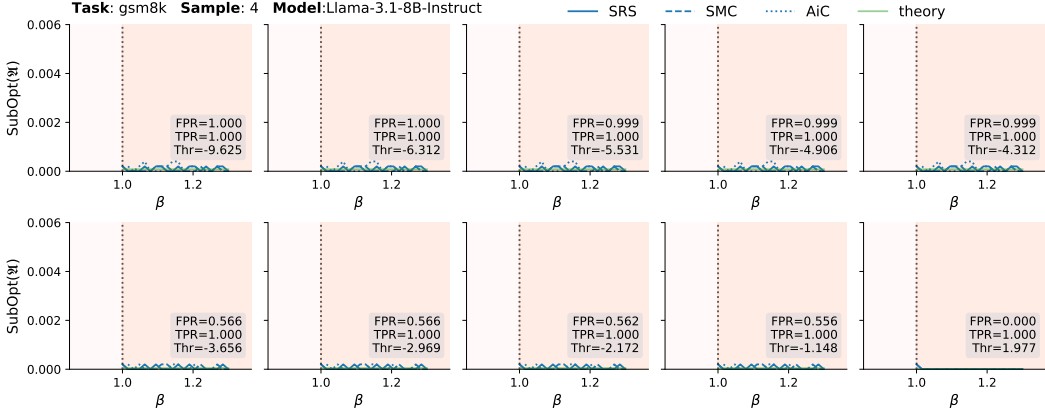

Figure 16: Sub-optimality as a function of $\beta$ for varying thresholds of the LLM-as-judge verifier, evaluated on sample 4 of the GSM8K dataset using `Llama-3.1-8B-instruct`.

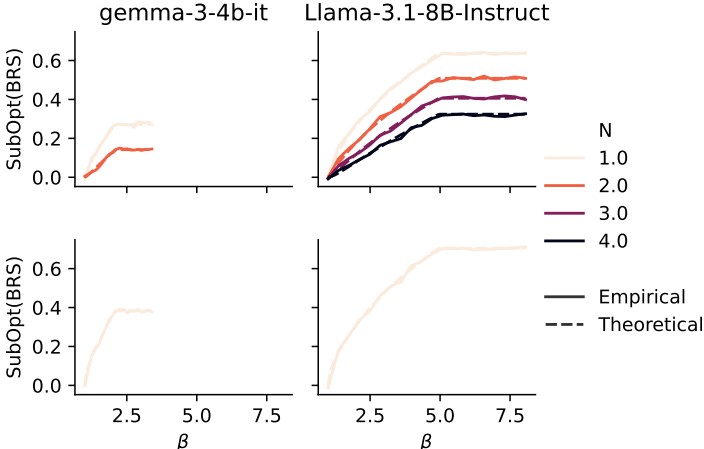

Figure 17: BRS plots with a **reward-guided verifier:** we plot **sub-optimality versus** $\beta$ with the ground truth verifier on the first row, and approximate verifier on the second row.

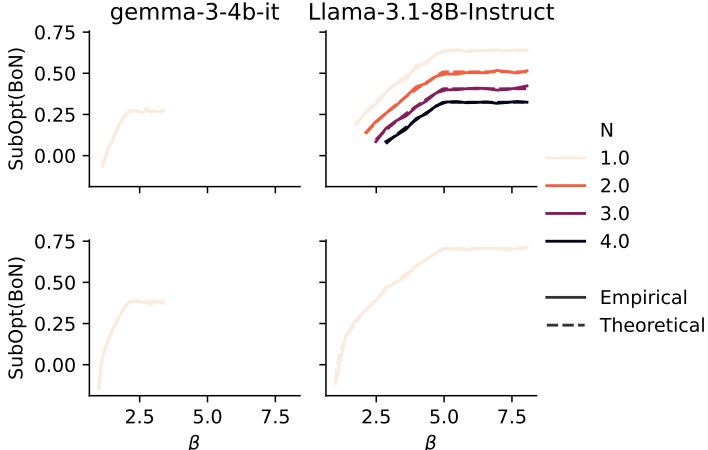

Figure 18: BoN plots with a **reward-guided verifier:** we plot **sub-optimality versus** $\beta$ with the ground truth verifier on the first row, and approximate verifier on the second row.

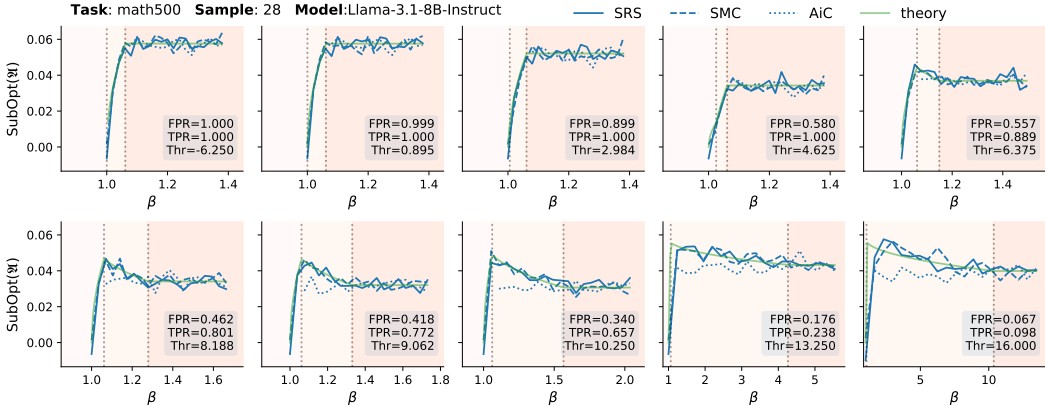

Figure 19: Sub-optimality as a function of $\beta$ for varying thresholds of the LLM-as-judge verifier, evaluated on sample 28 of the MATH500 dataset using `Llama-3.1-8B-instruct`.

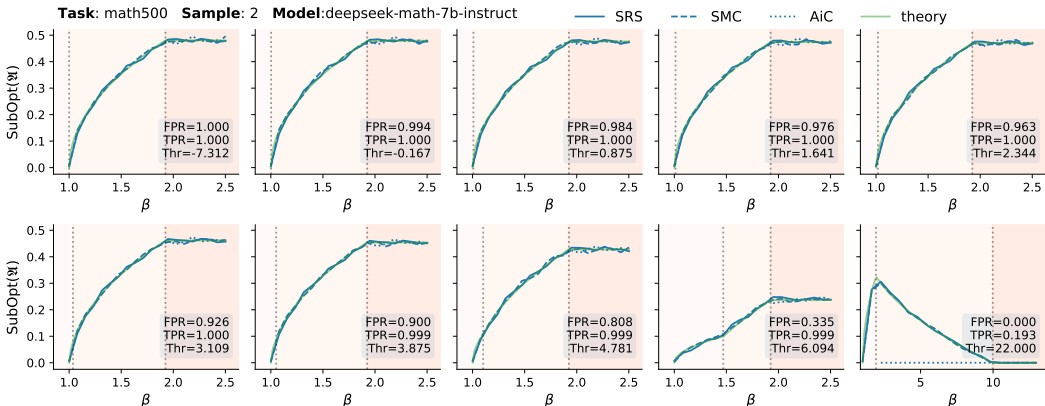

Figure 20: Sub-optimality as a function of $\beta$ for varying thresholds of the LLM-as-judge verifier, evaluated on sample 2 of the MATH500 dataset using `deepseek-math-7b-instruct`.

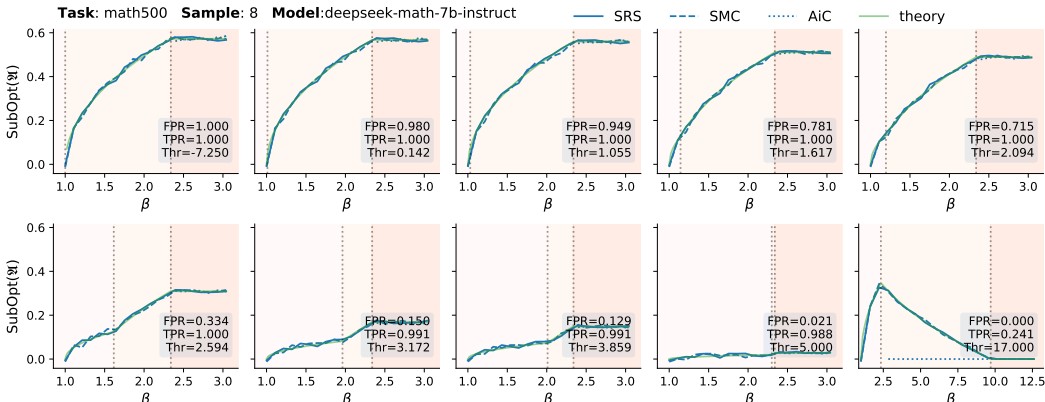

Figure 21: Sub-optimality as a function of $\beta$ for varying thresholds of the LLM-as-judge verifier, evaluated on sample 8 of the MATH500 dataset using `deepseek-math-7b-instruct`.

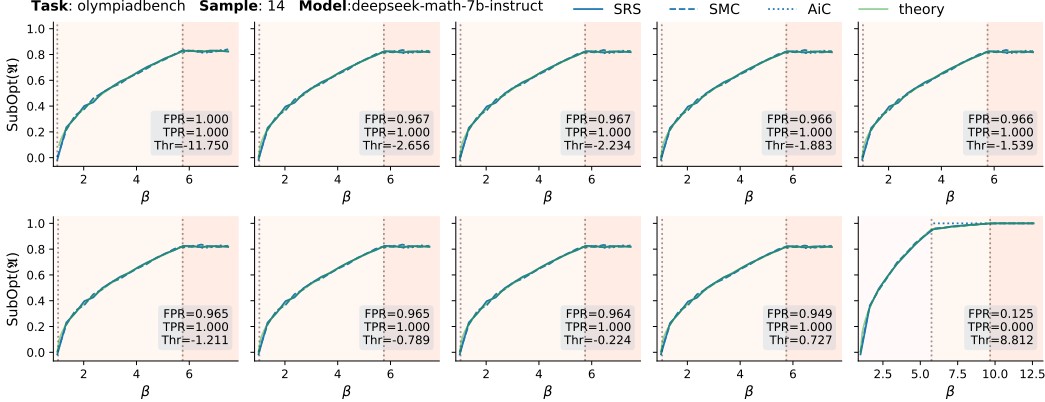

Figure 22: Sub-optimality as a function of $\beta$ for varying thresholds of the LLM-as-judge verifier, evaluated on sample 14 of the OlympiadBench dataset using `deepseek-math-7b-instruct`.

