# OpenReview forum: "Test-time Verification via Optimal Transport: Coverage, ROC, & Sub-optimality"
_ICLR.cc/2026/Conference — ICLR 2026 Poster_

### Official Review · Reviewer_NJAQ · 2025-10-27

**Soundness:** 4
**Presentation:** 4
**Contribution:** 3
**Rating:** 8
**Confidence:** 1

**Summary:**

This paper provides a framework for understanding test-time verification in LLMs by framing it as an optimal transport problem. The authors identify three components they consider important in the problem of verified generation:

(a) coverage, i.e. the generator's support over the solution space
(b) verifier ROC, i.e. the region of convergence and selectivity of the verifier
(c) sub-optimality, i.e. the algorithm's deviation from the optimal policy.

The authors model these factors as a transport problem and show three distinct "regimes": transport, policy improvement, and saturation. Then they analyze sequential and batch sampling algorithms and their computational trade-offs. The overarching goal of this paper is to understand and unify the empirical progress in test-time scaling on the one hand, and a mathematical framework describing how verification quality affects performance, on the other.

The authors show that verification helps only in certain regions of the coverage-suboptimality space. Concretely, in the policy improvement regime with moderate coverage, the verification yields the largest performance gains where it reduces sub-optimality.  The reason why verification helps is also analyzed.

That said, the practical implementation and the paper overall is outside of my area of expertise. I've alerted the ACs that I cannot judge this paper and giving it a confidence of 1.

**Strengths:**

See summary.

**Weaknesses:**

See summary.

**Questions:**

See summary.

---

> ### Author Response · Authors · 2025-11-23
>
> We thank the reviewer for taking the time to read our paper. Should you have any further question, we would be happy to clarify.

---

### Official Review · Reviewer_hmED · 2025-11-01

**Soundness:** 3
**Presentation:** 2
**Contribution:** 3
**Rating:** 4
**Confidence:** 3

**Summary:**

This paper studies the test time verification as a sampling problem with a lens of optimal transport. The framework of optimal transport where given a proposal distribution $\mu$, the goal is to sample from $\nu ^\star$ that is defined by $r ^\star$. The problem here is usually at test time, $r^\star$ is not available to guide the generation and thus one needs to rely on imperfect reward models. This paper also studies the geometry of sub optimality and coverage by relating the suboptimality to verifier's region of convergence (ROC) and generator's coverage, revealing three different regimes (transport regime, policy improvement regime and saturation regime). The paper analyzes: (a) sequential protocols—Accept-if-Correct (AiC), Sequential Rejection Sampling (SRS), Sequential Maximal Coupling (SMC); and (b) batched protocols—Best-of-N (BoN) and Batched Rejection Sampling (BRS). It proves AiC can violate the coverage constraint in low-coverage regimes and provides closed-form sub-optimality/complexity for SRS/SMC and exponential-in-N decay for BRS. Empirically, the paper focuses on GSM8K with Qwen, Llama and Gemma models and demonstrates their behaviour across different test time compute algorithms.

**Strengths:**

* Theoretical decomposition of sub-optimality into OTC is clear and a verifier-driven factor three-regime picture is intuitive and well explained.
* Precise AiC analysis with explicit coverage violation condition and sub-optimality formulas.
* SMC construction via maximal coupling with matching complexity to SRS; neatly links transport optimality with compute.
* BRS exponential improvement with tight envelope; theoretically beats BoN in low/intermediate regimes.
* Empirical studies match theoretical findings.

**Weaknesses:**

* GSM-8K is a very limited dataset, and especially as a test-time compute paper, I strongly recommend using a reasoning benchmark. You can either one or multiple of the AIME, OlympiadBench, MATH, GPQA or AMC questions to test the validity of the approaches here.
* The analysis of hybrid approaches between sequential and batched protocols such as beam search (also called as block-wise Best of N) is required.
* Actionable insights is not clear. How can the theoretical bounds and observations here help developers obtain state of the art performance across reasoning benchmarks?
* The narrative repeatedly generalizes that the verifier’s ROC “mediates” the coverage–sub-optimality trade-off, yet controlled experiments are limited to Qwen in the main text (Gemma/Llama only in appendix) and to one prompt. The strength of the claim should be toned down or supported with wider evidence.

Note: I am willing to update my score based on the rebuttal.

**Questions:**

* The acronym OTC and OHC both refer to optimal transport cost, right? Why does the paper use both interchangeably? I recommend sticking with one of them.
* The explicit verifier is constructed by ranking and selecting responses to hit preset $J$ and $s_{\text{ver}}$. Can you add experiments with more realistic verifiers (e.g., harness parser variants, unit-test fuzzing, LLM-judge with calibration) and show robustness of your theorems’ qualitative predictions under such noise?

---

> ### Author Response · Authors · 2025-11-23
>
> We thank the reviewer for the insightful questions; we provide a point-by-point answer to each question, and all responses to experiment-related queries can be found in the global response titled “global response to experiments”. We hope the reviewer reassesses the score if our responses are deemed satisfactory.
>
> **Comment:** *GSM-8K is a very limited dataset, … the validity of the approaches here.*
>
> **Response:** In the revised version, we include experiments on **three questions from MATH500 and one question from OlympiadBench (Appendix P)**. Across these problems we observe the same three-regime structure and similar relative behavior of sequential and batched methods, which supports the validity of our analysis beyond GSM8K, while a broader-scale evaluation on large reasoning suites is an interesting direction for future work.
>
> **Comment:** *The analysis of hybrid approaches … Best of N) is required.*
>
> **Response:** Thanks for the great suggestion. We would like to emphasize that this work focuses on *resampling*, and not *reasoning*—and hence, reward-guided (or verification-based) token-level decoding, possibly based on a “process reward model” is an important future direction, albeit, beyond the scope of our investigation. Analyzing beam search and its reward-guided variants (such as action-level rejection sampling) under the setting considered in this paper is a natural next step and demands an independent investigation.
>
> **Comment:** *Actionable insights is not clear...*
>
> **Response:** To summarize, quoting reviewer CAuT: “...actionable guidance: rejection sampling methods (SRS, BRS) are preferable under tight coverage constraints, while Best-of-N methods work better with relaxed coverage. This addresses a practical question practitioners face.” More specifically, under “low compute”, rejection sampling-style algorithms (SRS / BRS) is preferred, whereas, under “large compute”, best-of-N-style algorithms (AiC / BoN) are preferred. We explain this in the following.
>
> (i) **Low compute**: When compute (number of generator calls) is small, it implies a small coverage regime (since there is a one-to-one map between $\beta$ and compute), in which AiC and BoN can potentially have constraint violations. Hence, in this regime, SRS / SMC is preferred in the sequential setting, whereas BRS is preferred in the batched setting.
>
> (ii) **High compute regime:** In this regime, the sub-optimality and compute for all algorithms match; hence, AiC and BoN are preferred, since these do not depend on hyperparameters.
>
> **Comment:** *...controlled experiments are limited to Qwen in the main text (Gemma/Llama only in appendix) and to one prompt*
>
> **Response:** We have now added deepseek-ai/deepseek-math-7b-instruct as a generator and nvidia/AceMath-7B-RM as a reward model. Furthermore, we have added **three more prompts** from GSM8K—these can be found in Appendix P.
>
> **Comment:** *The acronym OTC and OHC both … with one of them*
>
> **Response:** Thanks for catching the typo—it should have been OTC throughout, we have now fixed this.
>
> **Comment:** *The explicit verifier is constructed by ranking and selecting responses...*
>
> **Response:** Please refer to the global response on experiments.

---

> ### Author Response · Authors · 2025-11-27
> **Gentle reminder: have we addressed your concerns?**
>
> Thanks, again, Reviewer hmED, for taking the time to review our paper. We would like to kindly follow up and check if our comments have addressed the reviewer’s questions and concerns.

---

### Official Review · Reviewer_CAuT · 2025-11-06

**Soundness:** 3
**Presentation:** 3
**Contribution:** 3
**Rating:** 6
**Confidence:** 4

**Summary:**

This paper develops a theoretical framework for test-time verification in LLMs using optimal transport theory. The authors characterize how three factors interact: generator coverage, verifier accuracy (region of convergence), and sampling algorithm sub-optimality. The main contribution is identifying three distinct regimes in the sub-optimality–coverage relationship: transport, policy improvement, and saturation. The paper analyzes both sequential (AiC, SRS, SMC) and batched (BoN, BRS) sampling algorithms, deriving exact bounds on computational complexity and sub-optimality. Experiments on GSM8K with Qwen, Llama, and Gemma models validate the theoretical predictions.

**Strengths:**

- The optimal transport formulation elegantly unifies the analysis of generator coverage, verifier imperfections, and sampling strategies. This provides a principled way to reason about test-time verification that goes beyond existing asymptotic analyses.

- The paper derives closed-form expressions for sub-optimality (Theorems 3.6, 3.8, 3.10) and computational complexity (Theorems 3.2, 3.5) across multiple algorithms. These exact results are more informative than asymptotic bounds for practical algorithm selection.

- Identifying transport, policy improvement, and saturation regimes is valuable both theoretically and practically. The analysis shows that the relationship between coverage and sub-optimality is non-monotonic and depends critically on verifier quality (Youden's index).

- Explicitly incorporating false positives and false negatives through ROC analysis (TPR, FPR) is important for practical deployment. Most prior work assumes perfect verifiers, which is rarely realistic.

- The paper provides actionable guidance: rejection sampling methods (SRS, BRS) are preferable under tight coverage constraints, while Best-of-N methods work better with relaxed coverage. This addresses a practical question practitioners face.

- Figures 4-5 show close alignment between theoretical predictions and empirical results across different models and parameter regimes, which strengthens confidence in the framework.

- The paper effectively uses figures (especially Figures 1-2), tables, and clear regime-based decomposition to communicate complex mathematical ideas.

**Weaknesses:**

- All experiments use one GSM8K question (sample 2), following Dorner et al. (2025). While this protocol has precedent, it limits our ability to assess whether the three-regime structure generalizes across problems of varying difficulty and characteristics. Even testing on 5-10 representative problems would strengthen the claims.

- SRS and SMC require knowing s_ver (the reference policy's mass on the verifier set), which may not be available at test time. Section 4 treats it as a hyperparameter, but Figure 5 shows that misspecification substantially degrades performance. The paper would benefit from a practical estimation procedure.

- The ℓ1-type constraint (Equation 1) based on χ²-divergence is mathematically convenient but somewhat restrictive. The authors discuss this limitation briefly but don't explore how alternative divergences (e.g., KL, Wasserstein) might change the regime structure. This seems like an important direction for future work.

- The experimental protocol samples with replacement from a pre-generated pool of 10,000 responses. This doesn't fully capture the parallel generation scenario that motivates batched methods in practice, where responses are generated independently in a single forward pass.

- While the paper analyzes expected number of proposals, wall-clock time depends on GPU parallelization, model size, and verifier overhead. Sequential methods may require fewer total proposals but could be slower than batched methods in practice. This practical tradeoff deserves discussion.

- The main experiments construct verifiers by controlling J and s_ver through probability-ranked selection. Real verifiers (unit tests, reward models) don't work this way. The reward-guided experiments (Appendix) are more realistic but limited to two models with FPR=0.3.

- The paper compares proposed algorithms against each other but not against recent test-time scaling methods like guided stream-of-search (Gandhi et al., 2024; Moon et al., 2024). Such comparisons would help position the work's practical contributions.

**Questions:**

- How robust is the three-regime structure across different GSM8K problems? Even a small-scale analysis (e.g., 10 problems stratified by difficulty) would be valuable.

- For practical deployment of SRS/SMC, what would you recommend for estimating s_ver? Could one use a small validation set, or are there theoretical estimation guarantees available?

- Does the Polish space formulation matter in practice? LLM outputs are discrete token sequences. Have you verified that discretization doesn't significantly affect the optimal transport characterization?

- How would the analysis change under alternative coverage constraints (KL-divergence, Rényi divergence)? Would the three regimes persist?

- Theorem 3.7 shows BoN violates coverage for β < s_r*(1-s_r*). Should practitioners avoid BoN in such regimes, or might constraint violation be acceptable if it improves empirical performance?

- How does performance vary across different FPR operating points in the reward-guided setting? Is there a principled way to select FPR?

- SMC and SRS achieve identical sub-optimality and complexity (Theorem 3.5). What's the practical advantage of the more complex SMC derivation?

- Could you clarify how your coverage parameter β relates to the sample budget N in prior BoN scaling analyses (Brown et al., 2024)?

- How might the framework extend to continuous reward models rather than binary verifiers? This seems relevant for many practical applications.

- Figure 5 shows SMC outperforming SRS when the assumed s exceeds the true value. Is there theoretical intuition for this crossover?

---

> ### Author Response · Authors · 2025-11-23
>
> We thank the reviewer for the valuable feedback. Response to queries about experiments can be found under the global comment “global response to experiments”. Here, we provided a point-by-point response to all other questions, and we hope the reviewer reassesses the score if our responses are deemed satisfactory.
>
> **Comment:** *All experiments use one GSM8K question (sample 2), following Dorner et al. (2025). While this protocol … representative problems would strengthen the claims:*
>
> **Response:** Thank you for pointing this out. Our theoretical three-regime structure is, in fact,**valid for all prompts**: Theorems 2.1 and 3.6 characterize the sub-optimality–coverage curve for *any* fixed prompt, and the three regimes (transport, policy improvement, saturation) arise solely from the coverage parameter $\beta$ and the verifier’s ROC. The specific GSM8K instance only affects constants such as $s_r^\star$ and Youden’s index $J$, which shift regime boundaries and saturation levels but do not change the qualitative three-regime geometry.
>
>
> That said, we agree that showing robustness across problems strengthens the empirical evidence. In the revision, we have **added experiments on three additional GSM8K test questions, one question from OlympiadBench, and three questions from MATH500**. For each of these prompts, we have plotted the sub-optimality–coverage curves in Appendix P.
>
> **Comment:** *SRS and SMC require knowing s_ver … estimation procedure:*
>
> **Response:** We agree with the reviewer about the necessity of a practical estimation procedure for $s_{\rm ver}$. In practice, $s_{\rm ver}$ can be estimated at a negligible cost using a held-out dataset. While $s_{\rm ver}$ is defined per prompt as the reference policy’s mass on the verifier’s set, practically, we may estimate a *global* $s_{\rm ver}$, whose sensitivity will depend on the distribution of the prompts.
>
> **Comment:** *The ℓ1-type constraint (Equation 1) … future work: (also – How would the analysis change … three regimes persist?)*
>
> **Response:** Thanks for the insightful suggestion. We answer this in two parts:
>
> i) Note that $\chi^2$ is a very “strict” divergence measure, in that it lower bounds most of the commonly used divergences [1]. To elaborate, for measures $\mu$ and $\nu$, let $D_2(\mu\|\nu)$ denote the second-order Rényi divergence. Then, we have $D_2(\mu\|\nu) = \log(1+\chi^2(\mu\|\nu))$ [1], which implies that the second-order Rényi divergence is upper-bounded by $\log(1+\beta)$, given a $\beta$-constraint on the $\chi^2$-divergence. Furthermore, for KL divergence, we have the relation $D_{\rm KL}(\mu\|\nu)\leq \log(1+\chi^2(\mu\|\nu))$ [1], which consequently enforces a $\log(1+\beta)$ constraint on the KL divergence. Finally, the 1-Wasserstein distance is equal to the total variation distance in our setting, and hence, using Pinsker’s inequality, we have that a $\beta$ bound on the $\chi^2$-divergence ensures that $W_1(\mu\|\nu)\leq\sqrt{0.5\log(1+\beta)}$. Hence, a $\chi^2$-constraint naturally ensures a constraint on various other divergence measures. This also implies that the three-regime structure persists for such other divergence measures.
>
> ii) Second, owing to the “strictness” of the $\chi^2$-divergence, it has been widely adopted in test-time scaling as well as post training investigations, see, e.g., Huang et al. 2025, Huang et al. 2024a, Huang et al. 2024b, Ji eta al. 2024.
> We have added a footnote in Section 2 summarizing this.
>
> Audrey Huang, Adam Block, Dylan J Foster, Dhruv Rohatgi, Cyril Zhang, Max Simchowitz, Jordan T Ash, and Akshay Krishnamurthy. Self-improvement in language models: The sharpening mechanism. arXiv:2412.01951, 2024a.
>
>
> Audrey Huang, Wenhao Zhan, Tengyang Xie, Jason D Lee, Wen Sun, Akshay Krishnamurthy, and Dylan J Foster. Correcting the mythos of kl-regularization: Direct alignment without overoptimization via chi-squared preference optimization. arXiv:2407.13399, 2024b.
>
> Xiang Ji, Sanjeev Kulkarni, Mengdi Wang, and Tengyang Xie. Self-play with adversarial critic: Provable and scalable offline alignment for language models. arXiv:2406.04274, 2024.
>
> [1] Nishiyama, T., & Sason, I. (2020). On relations between the relative entropy and χ 2-divergence, generalizations and applications. Entropy, 22(5), 563.
>
> **Comment:** *The experimental protocol samples … single forward pass:*
>
> **Response:** Indeed, we agree with the reviewer that sampling from a pool of $10,000$ responses does not perfectly capture sampling from the model in the batched scenario when one is interested in comparing the runtime of the algorithm. However, in this paper, we focus on the average number of proposals to represent the computational complexity. From a practical perspective, this methodology is adopted to speed up experiments, and is standard in recent literature on test-time scaling—see, e.g., Huang et al. 2025, Dorner et al. 2025.

---

> > ### Author Response · Authors · 2025-11-23
> >
> > **Comment:** *While the paper analyzes expected number of proposals, wall-clock time …  practical tradeoff deserves discussion:*
> >
> > **Response:** We agree and clarified this tradeoff. We use expected proposals as a hardware-agnostic complexity measure, but wall-clock time also depends on GPU parallelism, model size, and verifier cost, so batched methods can sometimes be faster despite using more proposals.
> > At the same time, our analysis shows that this comes with an important **accuracy–latency tradeoff**. Batched methods are inherently truncated (one or a few batches), and therefore **approximate** the optimal transport solution: they typically achieve **worse sub-optimality** than sequential methods run to comparable proposal budgets. Sequential methods can continue sampling until a high-quality solution is found and are therefore preferable when one prioritizes solution quality, operates under limited parallelism, or faces a heavy/serial verifier. In the revision, we have added a few sentences in **Appendix A** explicitly discussing this tradeoff between (i) lower wall-clock latency via batching and (ii) tighter sub-optimality guarantees from sequential procedures.
> >
> > **Comment:** *The paper compares proposed algorithms … (Gandhi et al., 2024; Moon et al., 2024). Such comparisons… practical contributions.*
> >
> > **Response:** Thanks for the great suggestion. We would like to emphasize that this work focuses on *resampling*, and not reasoning—hence, stream of search is not directly comparable. Specifically, we only assume access to a *verifier*, contrary to a *process reward model* for token level guidance. Indeed, we believe that the techniques developed in this paper should find relevance in the reasoning setting, and are a promising direction of future work.
> >
> > **Comment:** *Does the Polish space formulation … optimal transport characterization?*
> >
> > **Response:** The Polish space formulation readily extends to the discrete space formulation, where we change the integrals with summation. The discrete token space does not impact our results.*
> >
> > **Comment:** *Theorem 3.7 shows BoN violates coverage … empirical performance?*
> >
> > **Response:** This is an infeasible regime–theoretically, $\beta$ should always be more than 1, and hence this scenario would never arise in practice. We have modified this in the revised draft.
> >
> > **Comment:** *How does performance vary across different FPR … Is there a principled way to select FPR?*
> >
> > **Response:** Performance varies non-monotonically across FPR operating points in the reward-guided setting. If the verifier threshold is very low (high FPR), we effectively accept almost all responses, so the behavior matches the base generator. As we increase the threshold, more responses are rejected, and we move towards a regime where the accepted set overlaps maximally with the ground-truth verifier set, leading to improved sub-optimality. Beyond this point, further increasing the threshold over-prunes: we begin rejecting many good responses, overlap with the ground truth decreases, and sub-optimality may potentially worsen, eventually approaching the trivial reject-all behavior.
> > We have included an ablation in Appendix P that sweeps the verifier threshold and empirically illustrates this pattern. We observe (i) an initial regime where a weak verifier yields little or no policy improvement, and (ii) a “sweet spot” after which sub-optimality in the saturation region saturates as the threshold tightens. A principled way to select the FPR is therefore to choose the operating point that minimizes sub-optimality (or maximizes reward) in the saturation region on a held-out calibration set—i.e., the smallest FPR beyond which making the verifier stricter consistently degrades or saturates performance.
> >
> > **Comment:** *SMC and SRS achieve identical sub-optimality … more complex SMC derivation?*
> >
> > **Response:** The motivation of sequential maximal coupling (SMC) comes from the optimal transport viewpoint we propose, and the SMC sampling algorithm is directly obtained by minimizing the transport cost (i.e., forming an optimal transport plan). Presenting SMC serves two purposes:
> >
> > (i) SMC settles the natural curiosity, *what is the sampling algorithm obtained by optimizing the transport cost*? SMC serves as the answer to this question; furthermore, analyzing the computational complexity and sub-optimality of SMC, we observe that they match that of SRS–implying that an optimal transport map is as good as a valid transport map (SRS) in our setting.
> >
> > (ii) optimizing the transport cost in the case of autoregressive decoding gives *speculative decoding*, which offers up to 2x speed-up in decoding. From this insight, we speculate that given an additional (more powerful) model besides the draft model, SMC should outperform SRS.

---

> > > ### Author Response · Authors · 2025-11-23
> > >
> > > **Comment:** *Could you clarify how your coverage parameter β … BoN scaling analyses (Brown et al., 2024)?*
> > >
> > > **Response:** The $pass@k$ metric studied in Brown et al. (2024), in simple terms, quantifies the probability of generating at least one correct response from a pool of $k$ responses, *averaged over the dataset*. We view the $pass@N$ metric *per-prompt*, since our analyses adopt a per-prompt treatment. Specifically, we have the $pass@N$ metric as the probability of generating at least one correct response from a pool of $N$ responses (setting $k=1$) for a given prompt $\mathbf{x}$, which, in our verifiable setting, is given by:
> > >
> > > $$pass@N = 1-(1-s_{r^\star})^N$$.
> > >
> > > Furthermore, noting that $\chi^2(\nu_{\rm BoN}\|\mu) = \frac{1-s_{r^\star}}{s_{r^\star}} \cdot (1-(1-s_{r^\star})^N)$, and using the $\beta$ constraint on this divergence, we obtain that:
> > >
> > > $$\beta \geq \frac{1-s_{r^\star}}{s_{r^\star}}\cdot pass@N$$, (equation (28))
> > >
> > > which is the desired relation. Hence, in this setting, $\beta$ is proportional to the $pass@N$ metric. Indeed, Brown et al. (2024) calls the $pass@k$ metric the model’s “coverage”. Our coverage metric comes from an information-theoretic interpretation of the policy class.
> > >
> > > **Comment:** *How might the framework extend to continuous reward … practical applications*
> > >
> > > **Response:** Thank you for the great question. Going beyond verifiable rewards poses several non-trivial challenges, including (a) choosing an uncertainty metric for capturing reward estimation fidelity, (b) working out a closed-form for the likelihood ratio in the setting, and (b) analyzing the sub-optimality based on the transport framework we propose. While Huang et al. (2025) provides some results in the general reward setting, this is only limited to sub-optimality bounds and lacks a transport-oriented treatment. As such, we believe that this merits an independent investigation and is an important future research direction, which we state in our conclusions— “More broadly, moving beyond verifiable rewards toward general reward models for inference-time alignment is an important next step.”
> > >
> > > **Comment:** *Figure 5 shows SMC outperforming SRS … theoretical intuition for this crossover?*
> > >
> > > **Response:** Thanks for the sharp observation. The Radon-Nykodim derivative is a decreasing function in $s$, which implies that when we overestimate the verifier’s mass on the generator’s set $s_{\rm ver}$, it makes it harder for SMC to accept the first sample, pushing the algorithm to the AiC-type selection criterion (residual measure). The high reward of SMC is attributed to the AiC subroutine, albeit, at the cost of increased computation. On the other hand, this does not impact SRS because of the envelope scaling, which facilitates a faster acceptance criterion (smaller compute) at the cost of smaller average reward.

---

> > > > ### Author Response · Authors · 2025-11-27
> > > > **Gentle reminder: have we addressed your concerns?**
> > > >
> > > > Thanks, again, Reviewer CAuT, for taking the time to review our paper. We would like to kindly follow up and check if our comments have addressed the reviewer’s questions and concerns.

---

### Author Response · Authors · 2025-11-23
**Global Response to Experiments**

We sincerely thank all reviewers for taking the time to provide detailed comments on improving the paper, and for recognizing the merits of our work. We are thankful for the very insightful suggestions to improve our work. To address reviewers CAuT and hmED’s concerns about the experiments, we provide a point-by-point response to the suggested additional experiments below. For the remaining questions, we provided point-by-point answers to each reviewer.

Suggested additional experiments encompass the following dimensions (tagged with question number ordered starting from “1” corresponding to the reviewer):

- **More practical verifiers (CAuT Weaknesses-6, hmED Questions-2):** First, we would like to note that our first version already had LLM-judge verifiers using the Skywark reward model in Appendix P. We have added verifiers with varying false positive rates (FPRs) for the LLM-judge. These can also be found in Appendix P of the revised draft.

- **More datasets: (CAuT Weaknesses-1, Questions-6)** We have now included **three questions from MATH500 and one question from the OlympiadBench dataset** in Appendix P, which also showcase the three-regime structure predicted by theory.

- **More prompts and models: (CAuT Weaknesses-1, Questions-1, hmED Weaknesses-4)** As specified, we have now included **three prompts from MATH500 dataset, one from the OlympiadBench dataset, and three more questions from GSM8K**, further broadening the scope of our experiments. Additionally, we also include generations from deepseek-ai/deepseek-math-7b-instruct.These can be found in Appendix P.

---

### Meta-Review · Area_Chair_ZSra · 2026-01-14

**Summary:**

The paper studies the interplay among the generator's coverage, the verifier's region of convergence (ROC), and the algorithm's sub-optimality for test-time verification algorithms, including sequential scaling algorithms and batch scaling algorithms. The main contribution of the paper is to provide closed-form formula on different algorithms' sub-optimality as a function of the coverage constraint on the final obtained policy. The paper also conducts experiments to validate their derived formula.

During the discussion, the reviewers raised concerns on the practical insights obtained from the theoretical results, the generality of the derived regimes across different datasets and questions, how the result generalized to other coverage notions beyond $\ell_1$-type used in the analysis, the practicality of assuming $s_{ver}$ and other prompt/model-dependent quantities is known in practice. The authors addressed most of the concerns in the rebuttal.

While some of the concerns still exist, e.g., the practicality of assuming $s_{ver}$ and other prompt/model-dependent quantities is known in practice, I believe the paper makes a nice attempt at analyzing the test-time scaling algorithm through a principled lens, and hence the merit outweighs the flaw. I would recommend the paper for acceptance.

**Reviewer Concerns:**

See above discussion.

**Reviewer Scores:**

I expect Reviewer hmED might increase their score if sufficient discussion time is provided.

---

### Decision · Program_Chairs · 2026-01-26

Accept (Poster)